# SYNTHETIC BOOTSTRAPPED PRETRAINING

**Zitong Yang**[*,1,2], **Aonan Zhang**[*,1], **Hong Liu**[2]
**Tatsunori Hashimoto**[2], **Emmanuel Candès**[2], **Chong Wang**[1], **Ruoming Pang**[1]

[1]Apple  [2]Stanford University  [*]Equal contribution

## ABSTRACT

We introduce Synthetic Bootstrapped Pretraining (SBP), a language model (LM) pretraining procedure that first learns a model of relations between documents from the pretraining dataset and then leverages it to synthesize a vast new corpus for joint training. While the standard pretraining teaches LMs to learn causal correlations among tokens within a single document, it is not designed to efficiently model the rich, learnable *inter-document* correlations that can potentially lead to better performance. We validate SBP by designing a compute-matched pretraining setup and pretrain a 3B-parameter and a 6B-parameter model on up to 1T tokens from scratch. We find SBP consistently improves upon a strong repetition baseline and delivers up to 60% of performance improvement attainable by an oracle upper bound with access to 20x more unique data. Qualitative analysis reveals that the synthesized documents go beyond mere paraphrases – SBP first abstracts a core concept from the seed material and then crafts a new narration on top of it. Besides strong empirical performance, SBP admits a natural Bayesian interpretation: the synthesizer implicitly learns to abstract the latent concepts shared between related documents.

## 1 INTRODUCTION

Pretraining on the diverse internet texts is now seen to be bottlenecked by the rapid depletion of high-quality text data [64]. This imminent "scaling wall" motivates us to utilize existing data more effectively. Re-examining the conceptual foundation of pretraining, its success originates from the rich causal correlation among tokens *within* a document. However, this is not the only source of correlation pretraining datasets contain: a code document implementing the attention mechanism is derived from the arXiv preprint of the transformer paper; The book of Harry Potter is structurally similar to the screenplay of its movie production. Such connections suggest a weaker form of *inter-document* correlation derived from an underlying joint distribution of pretraining documents. We hypothesize that this additional signal, which is missed by standard pretraining, can be captured by synthetic data, presenting an underexplored avenue for improving performance.

To leverage this opportunity, we introduce Synthetic Bootstrapped Pretraining (SBP), a LM pretraining procedure that operates in three steps (Figure 1). First, SBP identifies semantically similar document pairs $(d_1, d_2)$, such as the transformer paper and its code implementation, from the pretraining dataset. Second, SBP models the conditional probability of $d_2$ given $d_1$, creating a "data synthesizer" that can synthesize a new, related document given a seed document. Finally, SBP applies the trained conditional synthesizer to the pretraining corpus itself, creating a vast text corpus that encodes the rich inter-document correlations that were previously missed (§2). By training a data synthesizer from the pretraining dataset itself, SBP avoids the pitfall of "bootstrapping" model performance using an external, readily available teacher LM, demonstrating a clean setup where the source of improvement stems from better utilization of the same pretraining corpus.

To test our hypothesis, we design a compute-matched, data-constrained experimental framework under which we pretrain a 3B-parameter and a 6B-parameter model on up to 1T tokens from scratch [34, 72], demonstrating the potential applicability of SBP for advancing frontier LMs. We compare SBP's performance against two crucial references: a strong repetition baseline, which represents the

---

Correspondence to `zitong@berkeley.edu`

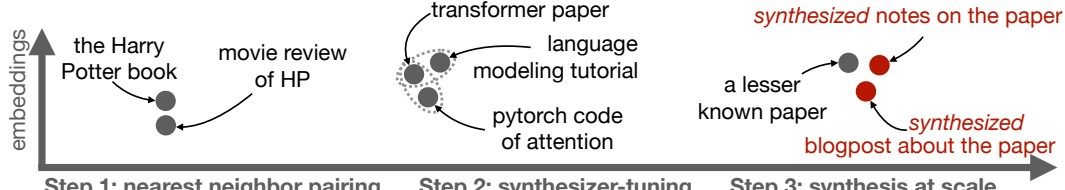

Figure 1: Data synthesis illustration of Synthetic Bootstrapped Pretraining (SBP): It first identifies semantically similar documents (**Step 1**) and then trains a conditional model that generates one element of the pair from the other (**Step 2**). Finally, SBP applies the conditional model to the pretraining corpus itself to synthesize a new, vast corpus for joint training (**Step 3**).

standard approach in data-constrained settings, and an oracle upper bound, which has access to an unlimited pool of unique internet data (§3). Our results show that SBP consistently surpasses the strong repetition baseline across different pretraining scales and closes up to 60% of the performance gap to the oracle with 20x additional unique data access (§4.1).

Besides strong benchmark performances, qualitative analysis of the synthesized documents reveals that they went beyond mere paraphrases of the real documents (§4.2). We postulate that the SBP synthesizer first abstracts latent concepts from the real document and then synthesizes a new document that expands upon the abstracted concept, incorporating diverse genres and content. We formalize this intuition through a Bayesian hierarchical concept model, where documents are related through shared concepts. From this perspective, we argue that the synthesizer implicitly learns a posterior likelihood model that abstracts latent concepts from the document – a mechanism not present in the standard LM pretraining (§5).

In summary, our contributions are threefold:

- **New pretraining framework:** We propose the Synthetic Bootstrapped Pretraining (SBP) algorithm that explicitly models inter-document correlations missed by standard pretraining practice and encodes those correlations into training via synthetic data.

- **Large-scale empirical validation:** We design a compute-matched pretraining setup that enables rigorous measurement of LM self-improvement and empirically validate SBP on 3B and 6B parameter models trained on up to 1T tokens from scratch.

- **Principled statistical interpretation:** We offer a natural Bayesian interpretation of SBP as implicitly learning a posterior for the latent concepts in a text document and concretize the intuition via qualitative analysis of synthesized documents.

In the remainder of the paper, we will first define the data-constrained pretraining problem we address and introduce the SBP technique we propose in §2. Then, we present the compute-matched experiment setup in §3 and results in §4. Finally, we conclude with a Bayesian interpretation of SBP that sheds light on the origin of the improved performance in §5.

## 1.1 RELATED WORK

Before we proceed, we review related work that highlights our contribution in three broad areas of research: LM pretraining, synthetic data for LM, and retrieval-augmented LM.

**LM pretraining.** The concept of pretraining, closest to its modern form, originates from a series of works including ELMo [47], ULMFiT [27], BERT [15], that propose to pretrain a neural network via an unsupervised objective and subsequently finetune for a wide range of downstream tasks. The GPT-series [48, 49, 8, 42] cemented the practice of using next-token prediction as the pretraining objective and applying it to large-scale crawled webpages as opposed to task-specific datasets (e.g., English-to-French translation). In recent years, the size of the pretraining corpora has grown rapidly, driven by the availability of massive web-crawled datasets, leading to a successful stream of dataset and pretrained model artifact: BERT [15, 37], GPT-2 WebText [49], CommonCrawl [12], CCNet [67], T5 C4 [51], the Pile [18], Gopher Massive Text [50], Llamda series [62, 16], Refined-Web [46], Dolma [56], DCLM-baseline [34], NemotronCC [57], etc. While pretraining has been tremendously successful, the rapid depletion of available internet text motivates us to shift our focus from acquiring more data to using the existing data more effectively.

**Synthetic data.** A natural way to overcome the limitations of scarce high-quality web data is to pretrain [20, 1, 2, 3, 61] or continually pretrain [69, 52, 71, 40] LMs on synthetic data. Existing approaches to data synthesis rely on distillation from a powerful "teacher" LM that generates compressed knowledge representation for the "student" LM to learn [26]. These teacher models must first undergo a human alignment process, which requires extensive human annotations and preference data [43]. Synthetic data from the teacher LM hints at a limited scaling trend: whilst the synthesized data from the teacher LM can be as impressive [13] as 7x more effective than real data, the performance improvement quickly converges to that of the teacher LM [9]. We instead consider the scenario where the sole source of world knowledge comes from a fixed set of pretraining documents (e.g., the internet) and algorithmically learn a data synthesizer with minimal human intervention (e.g., generative teacher models or human writing prompts). Therefore, our experiment setup simulates a situation where the LMs can self-boost their pretraining capability by refining their understanding of the fixed collection of pretraining documents.

**Retrieval augmented LM.** A natural class of methods that incorporates multiple documents together is retrieval augmented generation (RAG) [32, 33]. While originally introduced as a technique to be used at test-time for a domain-specific downstream task [6, 35], retrieval augmented approaches have been extended in scope: [30] and [68] implement RAG at pretraining scale and show improved test perplexity; [24] incorporates RAG at pretraining time by jointly training a retriever and the model itself for improved QA performance. [55] groups related documents into the same context window for improved long-context capability. In general, while the RAG-related approach enables the model to utilize rich inter-document correlations, it is fundamentally limited by the context window of the LM. In contrast, SBP encodes correlations into synthetic data that can be iteratively learned by the LM one document at a time. Prior to the advancement of embedding models that allow retrieving the entire document, [23] proposed retrieving neighboring pairs of sentences using Jaccard similarity and modeling the conditional distribution between them, similar to our conditional data synthesizer objective; however, they did not perform any pretraining experiments.

## 2 OUR METHOD

In this section, we introduce the data-constrained pretraining setup (§2.1) and then present the SBP procedure in three detailed steps (§2.2). We will present SBP as a general pretraining recipe by introducing a generic setup that includes a pretraining dataset, an LM architecture, and a collection of evaluation benchmarks. We defer the concrete compute-matched experiment design to §3.

### 2.1 DATA-CONTRAINED PRETRAINING SETUP

We consider a *data-constrained* setup where the goal is to train the best-performing LM given access to a fixed document collection $\mathcal{D}_{\text{pretrain}}$ (e.g., a snapshot of the entire internet). To establish a controlled experimental framework, we also choose a transformer architecture with parameters $\theta$ and a collection of held-out evaluation benchmarks $\text{Perf}$ (e.g., perplexity, few-shot QA accuracy). Recall that a transformer takes in a sequence of tokens and outputs a sequence of conditional probabilities of each token given all previous tokens. Applying the chain rule for joint probability, we can use a transformer to calculate the probability $p_\theta(y)$ of observing a particular text input $y$, or the conditional probability $p_\theta(y|x)$ of one piece of text $y$ followed by $x$.

Under such a setup defined by $(\mathcal{D}_{\text{pretrain}}, p_\theta, \text{Perf})$, pretraining searches for the best-performing transformer weights by maximizing the sum of the log-likelihood of pretraining documents $\arg\max_\theta \sum_{d \in \mathcal{D}_{\text{pretrain}}} \log p_\theta(d)$, and then evaluates the performance through $\text{Perf}(\theta)$. Statistically, this objective treats each document as an independent sample from a hypothetical distribution of all documents and attempts to learn this marginal distribution. However, this modeling assumption overlooks the structural similarities shared between natural language texts (e.g., Figure 1). We next present the SBP procedure that fills this gap.

### 2.2 SYNTHETIC BOOTSTRAPPED PRETRAINING

At a high level, SBP finds related document pairs $(d_1, d_2)$ from the pretraining dataset $\mathcal{D}_{\text{pretrain}}$ and trains a conditional synthesizer $p_\theta(d_2|d_1)$ using the same transformer architecture parametrized by

$\theta$. It then uses it to synthesize a large collection of documents $\mathcal{S}_{\text{pretrain}}$ to perform joint pretraining on $\{\mathcal{D}_{\text{pretrain}}, \mathcal{S}_{\text{pretrain}}\}$. The fact that SBP trains a data synthesizer from $\mathcal{D}_{\text{pretrain}}$ itself also distinguishes it from extensive existing work that relies on a readily available "teacher" LM.

**Step 1: Nearest neighbor pairing.** In preparation for training the conditional data synthesizer, SBP first curates pairs of related documents. To efficiently perform similarity search at pretraining scale, we adopt the Approximate Nearest Neighbor (ANN) methodology [38], which embeds each document as a quantized vector normalized to the unit sphere and then performs massively parallelizable linear algebraic operations. In our implementation of SBP, we use inner-product similarity, which we denote by $\langle d_1, d_2 \rangle$. Then, we select a subset of pairs whose similarity score exceeds a certain threshold $\alpha$: $\mathcal{D}_{\text{ST}} = \{(d_1, d_2) \in \mathcal{D}_{\text{pretrain}} \times \mathcal{D}_{\text{pretrain}}, \text{ s.t. } \langle d_1, d_2 \rangle > \alpha\}$. We provide the implementation details of paired data curation in §A.2.

**Step 2: Synthesizer-tuning.** SBP exploits the correlation between pairs of related documents by maximizing the conditional probability of $d_2$ given $d_1$: $\theta_{\text{ST}} = \arg\max_\theta \sum_{(d_1,d_2) \in \mathcal{D}_{\text{ST}}} \log p_\theta(d_2|d_1)$, which we obtain by summing over the log conditional probabilities corresponding to tokens from document $d_2$. We refer to this step as "synthesizer-tuning" as we are training a conditional probabilistic model that synthesizes a related $d_2$ from a given $d_1$. When performing synthesizer-tuning, we initialize $p_\theta$ at the transformer weights that has gone through normal pretraining. As a result, the model is equipped with the knowledge of individual documents at initialization, but not the conditional relation between them. Importantly, each document $d_1$ can be associated with multiple instances of $d_2$, encouraging the synthesizer to produce diverse, high-entropy outputs rather than deterministic synthesis.

**Step 3: Data synthesis at scale.** Finally, SBP synthesizes $\mathcal{S}_{\text{pretrain}}$ through a hierarchical sampling process: (I) First sample the seed document $d_1$ from $\mathcal{D}_{\text{pretrain}}$ uniformly at random; (II) Then sample synthesized document $d_2$ from $p_{\theta_{\text{ST}}}(\cdot|d_1)$. This process achieves synthetic data diversity utilizing two sources of variation: first through the variation of the seed documents $d_1$, which comes from the diversity of the pretraining document $\mathcal{D}_{\text{pretrain}}$ itself, and second through the entropy of the conditional distribution $p_{\theta_{\text{ST}}}(\cdot|d_1)$, which stems from the diverse inter-document correlations captured in $\mathcal{D}_{\text{ST}}$. While the procedure is empirically motivated, it actually admits a statistically principled Bayesian modeling of the distribution of natural language texts, which we explain in §5. For now, we focus on demonstrating the empirical effectiveness of SBP.

## 3 EXPERIMENT SETUP

In this section, we present our concrete experimental implementation of SBP. In a nutshell, we curated a pretraining dataset of 582M high-quality documents totaling 482B tokens from DCLM [34], designed 3B and 6B transformer architectures modified from the Llama 3 implementation [16], and chose nine commonly used benchmarks targeted at general world knowledge and commonsense reasoning (§A.1). We propose a *compute-matched* comparison scheme to validate SBP against natural reference methods at a compute scale of up to 1T total training tokens in our largest experiment (§3.1), bringing validation at a scale relevant for frontier LM development.

### 3.1 COMPUTE-MATCHED COMPARSION

We propose a *compute-matched* experimentation framework to rigorously compare SBP against two natural references: a repetition baseline where we repeat $\mathcal{D}_{\text{pretrain}}$ multiple times to utilize the available training compute and an oracle upper bound that enables the model to access as many unique documents as possible. Operationally, we control the training compute by controlling the total tokens seen during training, which is proportional to the training FLOPs given a fixed batch size and context window. We validate SBP across three different settings:

- **200B-scale**: In this setting, we cap the training compute to be 200B tokens and cap the data access at $\|\mathcal{D}_{\text{pretrain}}\| = 10\text{B}$ tokens.
- **1T-scale (3B)**: We also consider a larger scale closer to frontier model training, where we cap the training compute at 1T tokens and data access at $\|\mathcal{D}_{\text{pretrain}}\| = 50\text{B}$ tokens.

- **1T-scale (6B)**: To validate SBP on larger models, we additionally train a 6B-parameter model with the same 1T token budget and 50B unique data access.

For each training scale, $\mathcal{D}_{\text{pretrain}}$ with different sizes is sampled uniformly at random from the 582M documents pool. Given the compute-controlled comparison scheme, we next introduce two reference methods against which we compare SBP.

**Repetition baseline.** Since the compute budget typically exceeds the total number of unique tokens $\|\mathcal{D}_{\text{pretrain}}\|$, a natural baseline to use the additional compute is to repeat $\mathcal{D}_{\text{pretrain}}$ over multiple epochs. By design, in both 200B-scale and 1T-scale, we repeat the pretraining dataset $\mathcal{D}_{\text{pretrain}}$ 20 times to exploit the available compute budget. In practice, when the pretraining dataset comes from a mixture of different sources, higher-quality documents can be seen as many as 30 times during pretraining, while lower-quality texts may appear only once. [39] systematically evaluates the repetition baseline as a proposal to scale LMs under data constraints and finds that repeating $\mathcal{D}_{\text{pretrain}}$ up to 4 times yields nearly no performance degradation compared with having access to unlimited fresh data, but after around 40 times, repetition yields rapidly diminishing returns. Therefore, our choice of 20 times repetition with compute-matched comparison strikes a reasonable balance between efficient execution and maximizing performance gains from a fixed $\mathcal{D}_{\text{pretrain}}$ via repetition.

**Oracle upper bound.** Besides showing improvement against the repetition baseline, we also evaluate an oracle upper bound with unlimited data access. The motivation behind this is to contextualize the numerical improvement delivered by SBP. As we shall see in the next section, because different benchmarks respond differently to data size changes, SBP can deliver an improvement as large as 3.74% on some benchmarks but only 0.14% on others (Table 1). Also, as performance on LM benchmarks tend to scale logarithmically [44, 29] against data improvement, the numerical difference quickly caps out as we move from the 200B scale to the 1T-scale. By introducing this oracle upper bound, we can contrast the SBP improvement against this "oracle" improvement.

For the 200B-scale experiment, we implement the oracle upper bound as having access to 200B unique tokens from our document pool of size 482B tokens. For the 1T-scale experiment, we unfortunately do not have 1T unique documents due to the large fraction of duplicates from DCLM. As a surrogate, we utilize all 482B unique tokens as the dataset for training the oracle upper bound at the 1T-scale. We provide a partial justification for this by performing a scaled-down comparison at 400B training tokens, with one model having 400B unique tokens and the other one having 200B unique tokens repeated twice (§C.1). We find that the two models (400B unique and 200B repeated twice) yield nearly identical performance.

**Training recipe.** For both the repetition baseline and oracle upper bound at all scales, we use a batch size of 2,048 and a context window of 4,096, resulting in a throughput of 8M tokens per step. We apply a cosine learning rate scale with a 5% warmup to a peak learning rate of 1e-2, followed by subsequent decay to 5e-5 towards the end. Under this setup, pretraining costs 11K v5p-TPU hours at 200B-scale, 59K v5p-TPU hours at 1T-scale (3B), and 265K v5p-TPU hours at 1T-scale (6B). We adhere to this hyperparameter throughout the paper, including the SBP experiment presented next.

## 4 EXPERIMENT RESULTS

We perform SBP experiments under the compute-matched framework outlined in §3 at three levels of training compute budget: 200B-scale, 1T-scale (3B), and 1T-scale (6B). After joint training on real and synthetic data $\{\mathcal{D}_{\text{pretrain}}, \mathcal{S}_{\text{pretrain}}\}$, we find SBP consistently improves upon the repetition baseline across all scales (Table 1). In this section, we focus on presenting the performance of SBP and evaluating the quality of the synthesized pretraining data. We defer the implementation details of SBP to §A.2.

### 4.1 MAIN BENCHMARK PERFORMANCE

At the 200B-scale, we start with the source dataset of $\|\mathcal{D}_{\text{pretrain}}\| = 10$B and curate a SBP dataset of $\|\mathcal{S}_{\text{pretrain}}\| = 75$B tokens (detailed ablation in §A.3). We perform joint training on $\{\mathcal{D}_{\text{pretrain}}, \mathcal{S}_{\text{pretrain}}\}$ with the principle that we do not repeat any synthetic documents during training. This means that

Table 1: Computed-matched comparison of Synthetic Bootstrapped Pretraining (SBP) and oracle performance gains over the repetition baseline. On average, SBP delivers roughly 43% of the performance improvement in QA accuracy for the 3B model and 58% for the 6B model, attainable by an oracle with access to 20x more unique data.

| Benchmark | 200B-scale | | | 1T-scale (3B) | | | 1T-scale (6B) | | |
|---|---|---|---|---|---|---|---|---|---|
| | Baseline | SBP | Oracle | Baseline | SBP | Oracle | Baseline | SBP | Oracle |
| *Perplexity on held-out data ↓* | | | | | | | | | |
| OpenWebText2 | 5.74 | -0.53 | -1.02 | 4.51 | -0.02 | -0.12 | 4.25 | -0.06 | -0.21 |
| LAMBADA | 6.87 | -0.85 | -1.86 | 4.33 | -0.03 | -0.22 | 3.63 | -0.06 | -0.25 |
| Five-shot MMLU | 3.83 | -0.36 | -0.51 | 3.17 | -0.06 | -0.05 | 3.08 | -0.08 | -0.13 |
| *QA accuracy ↑* | | | | | | | | | |
| ARC-Challenge (0-shot) | 35.32 | +1.28 | +2.82 | 42.66 | +1.62 | +3.84 | 47.44 | +0.77 | +0.17 |
| ARC-Easy (0-shot) | 68.94 | +2.65 | +4.29 | 75.63 | +0.42 | +2.11 | 78.70 | +0.51 | +0.85 |
| SciQ (0-shot) | 90.50 | +1.00 | +2.40 | 93.20 | +0.80 | +0.50 | 92.90 | +1.90 | +1.80 |
| Winogrande (0-shot) | 60.14 | +1.90 | +5.53 | 65.19 | +1.42 | +2.92 | 70.17 | +0.47 | +2.36 |
| TriviaQA (1-shot) | 22.51 | +3.36 | +7.37 | 36.07 | +0.25 | +0.59 | 40.64 | +0.49 | +3.19 |
| WebQS (1-shot) | 8.56 | +3.74 | +10.83 | 19.34 | +0.54 | +0.44 | 19.88 | +3.79 | +5.22 |
| **Average QA accuracy** | **47.66** | **+2.32** | **+5.54** | **55.35** | **+0.84** | **+1.73** | **58.29** | **+1.32** | **+2.26** |

out of a 200B token training budget, we spent 37.5% of it on the 75B synthetic tokens from $\mathcal{S}_{\mathrm{pretrain}}$ without any repetition, and the remaining 62.5% on the real dataset $\mathcal{D}_{\mathrm{pretrain}}$ repeated 12.5 times. As shown in Table 1, SBP consistently decreases test loss and improves QA accuracy. On average, SBP captures $2.32/5.54 = 42\%$ of the improvement in QA accuracy delivered by the oracle run with 20x additional data access.

The training dynamics of SBP partly reveal its core mechanism. As we can see in Figure 2, initially, the baseline performs similarly to the oracle, since their training data share the same distribution, and when the number of tokens seen is small, there is no distinction between the two. Then gradually, the oracle becomes a better model than the baseline, as it has access to unlimited unique training data. For the SBP dynamics, it initially performs worse than both the baseline and the oracle, which is expected since the quality of the synthesized data at most matches that of the real data. However, gradually, SBP continues to scale while the baseline has plateaued. This suggests that $\mathcal{S}_{\mathrm{pretrain}}$ offers a signal $\mathcal{D}_{\mathrm{pretrain}}$ alone cannot capture.

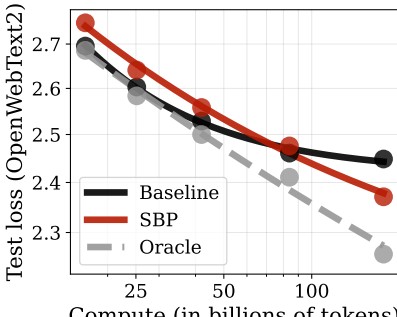

Figure 2: Training dynamics (200B-scale).

Lastly, to validate the benefit of SBP across different training scales, we implement a larger experiment with $\|\mathcal{D}_{\mathrm{pretrain}}\| = 50$B unique tokens under a compute budget of 1T total training tokens using both 3B and 6B-parameter models. Based on the ablation studies presented in §A.3, we include $\|\mathcal{S}_{\mathrm{pretrain}}\| = 125$B synthetic tokens for the 3B model and $\|\mathcal{S}_{\mathrm{pretrain}}\| = 250$B synthetic tokens for the 6B model, adhering to the principle of no repetition for synthetic data. Examining the results in Table 1, we observe that while perplexity-based measurements plateau for the 3B model [36], benchmarks like ARC-Challenge and Winogrande continue to show significant gains. On average, SBP recovers $0.84/1.73 = 48\%$ of the oracle's QA accuracy improvement for the 3B model. The results are even more pronounced for the 6B model, where SBP delivers a relative improvement of $1.32/2.26 = 58\%$ compared to the oracle. This demonstrates that SBP's effectiveness scales favorably with model size. The increased optimal synthetic data ratio for the 6B model suggests that larger models possess greater capacity to exploit the additional information in the synthetic corpus.

## 4.2 ANALYSIS OF SYNTHETIC DATA

In this section, we provide some qualitative and quantitative analyses of the synthesized documents to gain insight into the SBP procedure beyond what is measurable by the benchmark performance.

| Real document | Synthesis I | Synthesis II |
|---|---|---|
| HOME > Chowhound > California >

Coffeehouses in SD

I am always looking for good coffeehouses in San Diego. I am looking for coffeehouses with (obviously) excellent coffeebeans, baristas who know their business (good microfoam, good crema, they should know the different ratios for steamed milk, foamed milk, espresso for the cappuccino, latte etc., etc), good snacks (paninis, cakes etc), good (cold) flavored coffeedrinks and good ambience. So far I haven't found many who are good but would like to hear more suggestion.

Caffe Calabria: By far the best coffeehouse in SD. Excellent coffee and baristas (best cappuccino in SD,), great paninis. Only downside is that they close very early.

Café Lestat: Good example what happens with good coffee and average baristas. Lestat is also using coffeebeans from Caffé Calabria but it is sad to see that they don't know how to make excellent espresso, cappuccino.

E Street Café: Average espresso but strange flavored ... | Coffee = Love

On a recent trip to San Diego, one thing I wanted to find was coffee. I was worried it might be difficult with the huge decrease in Asian flavors, but I was pleasantly surprised. Many of the Mexican, Italian, and Cuban flavors that are pretty common in California are now in the mix. While I did find a few highly recommended spots, the majority of the locations I checked out were nothing to write home about. At the least, they were not being sold out, which I consider a plus.

One major factor that really needs to be taken into account is the type of machine being used. There are two types of espresso machines: the cheap ones that spit out lots of pressure and don't have any control over the shot, and the expensive ones that create amazing shots and utilize the types of superfine grinds you usually find in Italy. The latter types are much harder to find and cost significantly more.

Another factor to consider is the type of beans being used. If they are done correctly, great espresso can be made from any type of bean, but it's a rare experience to find a place that does ... | Discuss as:

Coffeehouses: San Diego. The best?

Café Lestat, by far, is a stand-out. (photo by Thomas Moriarty)

If the coffee scene in San Diego sounds like the one in New York, there's a reason. The upper echelons of NYC's drink scene are the same ones who have taken over the majority of the coffee houses in San Diego. This is probably partly due to the fact that both cities have large student populations, both are home to large coffee drinker bases, both have a large number of espresso shops, and both offer top-notch, handcrafted cappuccinos. But if there is one city in America that deserves to have its name above that of New York, it's San Diego.

There are just under 100 coffee shops in San Diego, with almost half of them located on University Ave. alone. So finding the perfect coffee shop is crucial. We spent a whole day just roaming around the area, hunting for the best.

In terms of the coffee itself, it's hard to beat Café Lestat. The baristas are amazing and their methods are pristine ... |

Figure 3: Comparison of original text with synthesized text variations.

**Qualitative examples.** We first show some samples of synthesized documents from the 200B-scale experiment (Figure 3) with more samples from 1T-scale (3B) presented in §B.4. On the left, we display a real document about a practical, first-person guide to the coffee houses in San Diego. Then, we present two synthesized texts that exhibit notable differences in both framing and depth, with varying degrees of fidelity to the seed document. Synthesis I sticks to the same topic but shifts toward an expository essay on espresso machines and bean quality, with little mention of specific coffee shops. Synthesis II adopts a promotional, comparative style, linking San Diego's coffee culture to New York's and praising Café Lestat in a way that departs from the original's balanced assessments. Note that SBP provides no instructions on how the synthesizer should use the seed texts to write new documents. The model spontaneously learns to introduce new content and style into the discussion while staying on topic. In fact, it is challenging to manually craft a prompt to an instruction-tuned model that would output either Synthesis I or II with the real document as input. This example highlights how SBP differs from existing paradigms of data synthesis – the output somehow first abstracts the seed document and then synthesizes new text with more generalized narratives, genres, and intent. We provide more extensive analysis of this observation in §5.

**Quantitative Analysis.** In addition to qualitative examples, we also conduct quantitative evaluations to assess the quality of the generated texts. We measure text distributions for the synthesized document at 200B-scale and 1T-scale. To establish a reference, we also conduct the same evaluation on the real documents. We measure five basic quality indicators:

- **Repetition:** A document may contain too many repeated sentences or patterns. Repetition rate refers to the fraction of documents that exhibit this problematic behavior.

- **Duplicate@1M:** Another failure mode of synthesis is when the documents sampled from the synthesizer distribution are nearly duplicates of each other. Duplicate@1M refers to the fraction of unique documents (determined by Jaccard similarity at a threshold of 0.6) when 1M documents are sampled from the text distribution.

- **Non-factual:** A common failure mode of synthesis is the generation of content that contradicts established knowledge or facts. Non-factual rate refers to the fraction of documents that contain verifiable factual errors, as determined by automated fact-checking tools.

- **Pair-irrelevance:** The synthesized $d_2$ is considered relevant to $d_1$ if they pertain to the same topic, event, entity, person, place, or object. Pair-irrelevance refers to the fraction of synthesized $d_2$ that is not relevant to $d_1$, indicating the synthesis is not rightly using information from $d_1$.

- **Pair-copying:** $d_1$ and $d_2$ are considered near-duplicates if they are almost identical, except for some extra white spaces, line breaks, or punctuation. Pair-copying refers to the fraction of synthesized $d_2$ that is a near duplicate of $d_1$.

Operationally, we implement Repetition, Pair-irrelevance, and Pair-copying using LM-as-judge (prompts and more implementation details given in §B.3) by sampling 1,000 examples from each distribution and estimating the fraction of documents satisfying each criterion. For Non-factual (prompts and details given in §B.2), we sample 10,000 examples and conduct a comprehensive examination of factual errors to ensure broader coverage of the generated data. For Duplicate@1M, we use rule-based filtering to detect the fraction of duplicates based on 1M documents sampled from each distribution. We present the result in the table below. All metrics are lower for better data.

Table 2: Quantitative evaluation of documents sampled from the synthesizer at 200B-scale and 1T-scale. We can see that the synthesized documents preserve topics and are not are simple duplicates.

|  | Repetition ↓ | Duplicate@1M ↓ | Non-factual ↓ | Pair-irrelevance ↓ | Pair-copying ↓ |
|---|---|---|---|---|---|
| **200B-scale** | 4.3% | 0.8% | 15.1% | 25.6% | 0.1% |
| **1T-scale (3B)** | 3.9% | 0.8% | 8.7% | 7.8% | 0.9% |
| **1T-scale (6B)** | 2.6% | 0.3% | 6.5% | 6.0% | 0.3% |
| **Real data** | 1.8% | 0.7% | 1.8% | n.a. | n.a. |

At a high level, Repetition and Duplicate@1M measure a basic text quality that is independent of the specific pair-synthesis strategy employed by SBP. They aim to detect two simple failure modes: text repetition, a common failure pattern in generations from small language models (3B in our case), and the lack of diversity, a common issue with synthetic data that relies on variation induced by the sampling temperature. From Table 2, we find that both 200B-scale and 1T-scale synthesis match the quality of real data as captured by these two metrics. We note that the absence of repetitions and duplicates is not, in itself, an indicator of high-quality or educational text, but rather a basic sanity check that ensures the synthesized texts are diverse.

Non-factual failure stems from hallucinations that introduce non-existent entities or relations inconsistent with reality. We find that synthesis at the 1T-scale (3B) significantly reduces these errors compared to the 200B-scale. Furthermore, with the 6B-parameter model, the non-factual rate further decreases to 6.5%. This indicates that as the data synthesizer is trained on more data and with larger models, the factuality of the generated outputs tends to converge toward that of real data.

Pair-irrelevance and Pair-copying, on the other hand, measure how synthesized $d_2$ relates to the seed $d_1$. There are two failure modes we would like to detect: first, when $d_2$ is completely irrelevant to $d_1$, and second, when $d_2$ merely copies the content of $d_1$. We observe that both 200B-scale and 1T-scale synthesis avoid simply copying and pasting $d_1$. More interestingly, we observe that the 1T-scale demonstrates substantially higher relevance than the 200B-scale, which intuitively makes sense as the synthesizer learns more diverse relations among $|\mathcal{D}_{\text{pretrain}}|$ =60M documents than $|\mathcal{D}_{\text{pretrain}}|$ =12M corpus.

At this point, we have shared the main results of the experiment. In the appendix, we present the implementation details of SBP in §A.2, ablations involving synthetic data mixture ratio in §A.3, additional analysis of synthesized documents in §B, and comparsion with using a larger 6B model in §C.2.

## 5 STATISTICAL FOUNDATIONS OF SBP

In this section, we present a Bayesian interpretation of the SBP procedure, offering one potential explanation for the origin of the SBP improvement. We will formulate a hierarchical model of natural language texts (§5.1) and demonstrate that SBP implicitly enables LMs to learn a posterior standard pretraining cannot capture. We conclude by connecting our findings from this idealized model to the reality of LM (§5.2). We begin with the observation that the pretraining objective models the marginal likelihood of documents:

$$\arg\max_{\theta} \log p_\theta(\mathcal{D}_{\text{pretrain}}) = \arg\max_{\theta} \sum_{d \in \mathcal{D}_{\text{pretrain}}} \log p_\theta(d). \tag{1}$$

However, different natural language documents share structural similarities (Figure 1), which suggests a potentially more complex underlying joint distribution that we will explore next.

## 5.1 A HIERARCHICAL CONCEPT MODEL FOR NATURAL LANGUAGE

In the transformer example from Figure 1, both the arXiv preprint of the transformer paper and its code implementation are derived from the abstract concept of "transformer neural network". From this perspective, we can view the generation process of natural language documents as a hierarchical sampling process where we first sample a collection of abstract concepts $c^{(i)}$ (e.g., the idea of a transformer) from a semantic space of all concepts $\mathcal{C}$ and then generate new documents $d^{(i,j)}$ conditional on $c^{(i)}$. If we adopt this view, we can think of the pretraining document as follows.

- **Concept sampling**: Sample a fixed concept collection $\{c^{(i)}\}_i \sim P(c)$.
- **Document generation**: For each concept $c^{(i)}$, generate docuemnts from $\{d^{(i,j)}\}_j \sim P(d|c^{(i)})$ constituting one part of the pretraining dataset.

Under such a model, the structural similarity between documents generated from the same concept is modeled as probabilistic *dependence*. The standard pretraining objective (1) then neglects inter-document correlation and only learns the marginal distribution $P(d) = \int_{c \in \mathcal{C}} P(d|c)P(c)dc$. In this view, the model learns to generate plausible text by first generating a core concept $c$ and then performing the generation $P(d|c)$. In contrast, the synthesizer-tuning objective models a posterior of $c$ given $d$. To see this, we additionally assume that the curated pairs $(d_1, d_2)$ come from the same underlying concept $c$. Then, the synthesizer-tuning objective (§A.2) forces the LM to perform a distinct task: $P(d_2|d_1) = \int_{c \in \mathcal{C}} P(d_2|c)P(c|d_1)dc$. Here, we use Bayes' rule and the conditional independence assumption $P(d_2|c, d_1) = P(d_2|c)$, which says that the documents from the same concept are conditionally independent given that concept. As a result, to successfully model $P(d_2|d_1)$, the synthesizer must first perform posterior inference to infer the latent concept $c$ given the document $d_1$, and then use this inferred concept to synthesize a new document $d_2$, a signal that is ignored by the standard pretraining objective. To illustrate this concretely, we perform a post-hoc analysis by prompting an LM to identify the shared concepts between the synthesized document and its seed (Table 4 in §B). We can see that while it is difficult to describe a synthesized document as the outcome of a simple transform, such as a paraphrase or summarization, it always share a common underlying concept with its seed origin.

The additional signal from the posterior then enables a form of self-distillation. The synthesizer, by learning a more complex conditional objective, becomes a more knowledgeable "teacher" model that has learned to infer the latent structure of data. The synthetic data it produces is then the knowledge "distilled" from this teacher [26]. The final LM training acts as a "student" learning from real and synthetic data, enabling it to uncover information that real data alone cannot reveal.

## 5.2 FROM IDEALIZED MODELS TO LANGUAGE MODEL REALITY

For real text documents, we do not know the true data-generating process, and any parametric assumption would be incorrect. This is where the power of the transformer neural network shines. A transformer is a *mapping-first* [7] approach. It does not require explicit modeling of the underlying parametric model. Instead, as a universal function approximator [10], it directly learns the complex conditional distribution $p_\theta(d_2|d_1)$ from paired data alone.

In this context, the transformer's ignorance of an explicit hierarchical model is its blessing. It bypasses the impossible step of modeling the true hierarchical distribution of language and instead brute-forces the learning of the exact transformation required: the end-to-end process of posterior inference and subsequent synthesis. The self-distillation framework – synthesizing data from this conditional model and then training on it – is all that is needed. We never need to introduce an explicit hierarchical model to perform the forward $P(d|c)$ and backward pass $P(c|d)$ in the latent space. The entire procedure is implicitly carried through the synthesizer-tuning update with the latent concept $c$ integrated, demonstrating a powerful insight for scaling LMs in the real world.

## 6 DISCUSSION

Before the prevalence of pretraining [49, 15], we would need 40M pairs of English and French sentences [60] to grant LM the ability to translate from one language to another. In contrast, any modern LM [19, 42, 21] can easily achieve this task via a single prompt. Such impressive advancement owes its origin to the rich correlations between words within a document LMs have learned during pretraining. This shift from a hard-to-collect, task-specific dataset to the readily available, generically

crawled internet data marks a transition from relying on strong but scarce supervision to a weak self-supervision that is easy to scale. As we gradually deplete this weak source of self-supervision by exhausting the available internet data, many have called for stronger forms of supervision, such as reinforcement learning [14, 41]. We instead offer an alternative perspective that continues to search for a form of self-supervision weaker than next-token prediction. SBP offers a particular instantiation of such an effort by examining the correlations *between* documents missed by the current pretraining paradigm. It remains to explore other forms of supervision not currently utilized.

The fact that SBP could provide any improvement originates from the poor inductive bias of the transformer neural network. For example, transformers trained on the text "A is B" can not generalize to "B is A" [5], requiring the user to curate synthetic data that explicitly narrates the converse relation [69]. Similarly, one can imagine an architecture with very good inductive bias such that the LM trained individually on $d_1$ and $d_2$ can automatically internalize the relation between the two. Despite such poor inductive bias, transformers have the impressive advantage of being more parallelizable and scalable than their alternatives [63, 54]. Given this trade-off, SBP offers a unique solution that preserves the system benefits of the transformer architecture while also enabling the learning of missed correlations by encoding such additional signals via synthetic data.

## 6.1 FUTURE DIRECTIONS

**Document embedding with activations of pretrained LM**   In our implementation of SBP, we use Qwen3-0.6B-Embedding [70] to obtain embeddings of DCLM [34] documents. An ideal implementation of SBP would only rely on the 3B-parameter model and the pretraining dataset itself to curate the paired synthesizer-tuning dataset. To achieve this, we can use the activations of the self-attention layer from an intermediate transformer block as a learned representation of documents. [30] and [68] implemented this at the much smaller scale of $\sim 300$M parameters and $\sim 3$B tokens. However, our experiments operate at a much larger scale with a customized model. As a result, we utilize the optimized vLLM [31] inference infrastructure for Qwen3-0.6B embedding models to efficiently index the pretraining corpus. Since the SBP procedure only requires a coarse binary decision of relevant vs. not relevant, which is much weaker than fine-grained document ranking embedding models are optimized for, we leave the more involved inference infrastructure for future work.

**Parametric fit of SBP scaling law**   One experimental consequence of LM pretraining is a clean scaling law [29, Equation 1.4] that relates the held-out test loss $L(N, D)$ to the number of LM parameters $N$ and the size of the pretraining dataset $D$. A natural scientific question is how the scaling law of SBP compares to the scaling law of pretraining. In our experiments, we evaluate $L(N, D)$ at three distinct points: $(N = 3\text{B}, D = 10\text{B})$, $(N = 3\text{B}, D = 50\text{B})$, and $(N = 6\text{B}, D = 50\text{B})$. We observe that SBP delivers larger relative improvements with the 6B model, suggesting a favorable scaling behavior. There are two obstacles to a full scaling law: first, SBP is inherently a large-scale algorithm that cannot be scaled down. Since SBP first uses the pretrained LM to generate synthetic text and then trains on it, if the model and dataset sizes are too small, the generated text may not even be coherent. In contrast, the scaling experiments in [29] involve model sizes ranging from 768M to 1.5B and dataset sizes ranging from 22M to 23B, which allows for efficient experimentation. Second, varying $N$ or $D$ implies redoing the synthesizer-tuning and subsequent data synthesis over billions of tokens. Additionally, varying $D$ also implies redoing the nearest neighbor matching, as the neighbors are only defined given a fixed document pool. Obstacles aside, it would be interesting to see whether the SBP scaling law differs from the normal scaling law by a smaller multiplicative factor or a better exponent. Since SBP additionally utilizes inter-document correlations, a form of long-range interactions, its scaling law not only helps us understand SBP but also potentially helps us better understand natural language data itself [17].

## 6.2 CONCLUSION

In conclusion, we introduce Synthetic Bootstrapped Pretraining (SBP) as a new LM pretraining framework that leverages inter-document correlations missed by the standard pretraining objective. We demonstrate the effectiveness of SBP by pretraining 3B and 6B-parameter models from scratch for up to 1T tokens under a rigorously designed compute-matched experiment setup. Qualitative and quantitative analyses of the synthesized text reveal rich variation that cannot be captured by simple paraphrases. SBP offers a natural Bayesian interpretation, implicitly learning a posterior that infers the latent concept in a document while being practically effective. Its *bootstrapping* nature grants SBP the possibility of scaling the pretraining capability of the LM beyond its current limit.

## ACKNOWLEDGEMENT

We would like to thank Zonglin Li for guidance on the implementation of ScaNN, Percy Liang for discussion regarding the curation of paired data, and Andrew Ilyas, Bailin Wang, Barry Theobald, Xiang Kong, Xin Cheng, Yizhe Zhang, Zhiyun Lu for their feedback on the early draft of this paper. ZY was supported by the Albion Walter Hewlett Stanford Graduate Fellowship. EC was supported by the Office of Naval Research grant N00014-24-1-2305, the National Science Foundation grant DMS-2032014, and the Simons Foundation under award 814641. TH was supported by a grant by HAI, DSO labs, gifts from Schmidt Sciences and a grant under the NSF CAREER IIS-2338866, ONR N00014-24-1-2609, and DARPA Cooperative Agreement HR00112520013. This work does not necessarily reflect the position or policy of the government, and no official endorsement should be inferred.

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

## CONTENTS

# A   ADDITIONAL DETAILS ON SYNTHETIC BOOTSTRAPPED PRETRAINING

## A.1   DATA, MODEL, AND EVALUATION

In this section, we present the complete details of the experiment set referenced in §3.

**Dataset.**   A typical pretraining dataset is a mixture of different sources (e.g., GitHub, arXiv, CommonCrawl, etc.) with distinct sampling weights assigned to each constituent. We simplify this reality by considering a fixed document collection, which is a customized version of the DCLM dataset [34]. The original 4T token DCLM-baseline split contains roughly 80% duplicates, as reported by [72]. Therefore, we begin with the de-duplicated dataset, which consists of 769B tokens. We clean the raw Zyphra de-duplicated data by normalizing repeated line breaks, removing long URL links, and fixing malformed Unicode characters. For efficiency reasons, we cap the context window of the synthesizer-tuning (§3) step at 8,192 tokens. As a result, we additionally filter out the documents whose length is above 4,096 tokens, allowing both $d_1$ and $d_2$ to fit into the context window in the worst case when both documents are 4,096 tokens long. After all the de-duplication, cleaning, and filtering procedures, we end up with a collection of 582M high-quality documents $\mathcal{D}_{\mathrm{pretrain}}$ totaling 482B tokens. We use the notation $|\mathcal{D}_{\mathrm{pretrain}}|$ to denote the number of documents in the pretraining dataset and $\|\mathcal{D}_{\mathrm{pretrain}}\|$ to denote the total number of tokens.

**Architecture.**   We use the Llama 3 transformer architecture [16] to model the probability $p_\theta$ with the notable exception of implementing a QK-norm on top of the existing design, which we empirically find to stabilize training. Our resulting model is a 3B-parameter 26-layer transformer model with a hidden dimension of 3,072. Each layer employs grouped query attention with 24 query heads and 8 key/value heads. To validate the scalability of SBP, we also train a 6B-parameter model with 32 layers, a hidden dimension of 4,096, 32 query heads, and a feedforward dimension of 13,056 (detailed in Table 12). The position embedding is RoPE [58] for queries and keys, with frequency 5e+5. The feedforward network (FFN) has hidden dimension 8,064, and we apply prenorm to both the attention and FFN blocks. For tokenization, we implement a customized BPE tokenization with a vocabulary size of 49,152. To match the 8,192 context window design for synthesizer-tuning we have mentioned, we use context window 4,096 for pretraining, so that every document in $\mathcal{D}_{\mathrm{pretrain}}$ can fit into the context window.

**Benchmarks.**   To assess the pretraining capability of LM, we measure pretraining test loss and general world knowledge benchmarks. We evaluate held-out test perplexity (exponential of negative log-probability) on 1) OpenWebText2 from EleutherAI [49]; 2) Narrative understanding with LAMBADA [45] and 3) Broad domain multiple-choice with MMLU [25]. We evaluate QA accuracy on 4) Hard scientific reasoning with ARC-Challenge [11]; 5) Easy scientific reasoning with ARC-Easy [11]; 6) Scientific QA with SciQ [66]; 7) Common sense reasoning with Winogrande [53]; 8) Reading comprehension with TriviaQA [28]; 9) Openbook QA with WebQS [4]. We directly evaluate the pretrained model with either zero-shot or few-shot prompts. Although MMLU is more commonly known as a QA benchmark, we find that evaluating MMLU accuracy for weak models yields a highly non-smooth readout. As a result, for each MMLU test question, we prepend the question with a 5-shot example of QA pairs and postpend it with the correct answer. Then, we treat each such sample as a text corpus and evaluate LM's perplexity on such a text sample. Empirically, we find that this perplexity-based MMLU correlates well with MMLU accuracy when the underlying model is large enough to yield a stable readout, and also delivers smooth performance changes for smaller models. Note that those benchmarks are known to improve significantly with instruction finetuning [65]. However, we stick to our data-constrained setup and do not introduce any additional data that may confound the comparison.

## A.2   SBP IMPLEMENTATION DETAILS

In this section, we present the implementation details of SBP outlined in §2.

**Nearest neighbor pairing**   Recall from §3 that we work with 3B and 6B-parameter transformer architectures and pretraining dataset at $\|\mathcal{D}_{\mathrm{pretrain}}\|$ =10B and $\|\mathcal{D}_{\mathrm{pretrain}}\|$ =50B scale. To take advantage of efficient ANN search at pretraining scale, we embed the documents from $\mathcal{D}_{\mathrm{pretrain}}$ as 1,024

dimensional vectors using Qwen3-Embedding-0.6B. Then, we use ScaNN [22, 59] with 8-bit quantization to perform efficient similarity search. We adopt an asymmetric sharding to keys and value vectors. For each value vector, we build a ScaNN search tree with $\sqrt{N}$ leaves where $N$ is the number of vectors in each value shard. To distribute the key shards across each search tree, we employ a "salting" strategy, where we create multiple copies of the ScaNN searcher and assign one key shard to each salted copy of the searcher (Figure 4). This design enables us to perform a top-200 nearest neighbor search over $|\mathcal{D}_{\mathrm{pretrain}}| =$ 60M documents within 155M CPU hours.

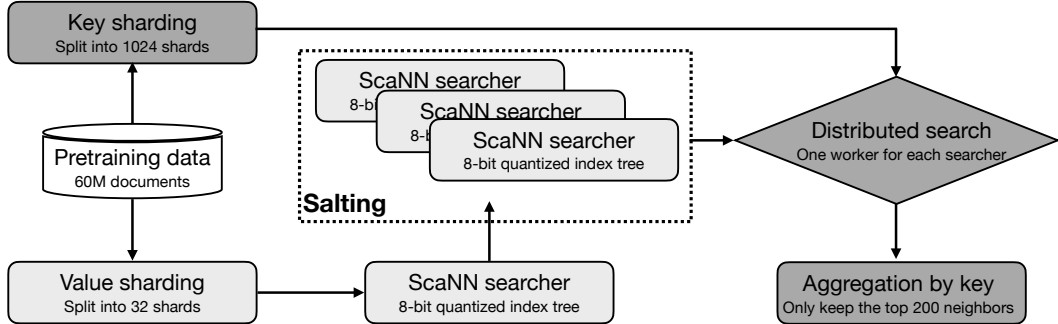

Figure 4: ScaNN system design for efficient distributed search.

At all scales, after obtaining the top 200 neighbors for each sample, we select the pairs whose similarity score is greater than 0.75. We chose this cut-off as it would later lead to a tractable size of synthesizer-tuning dataset $\mathcal{D}_{\mathrm{ST}}$. To access the effect of choosing a different threshold, we provide a quantitative analysis of the fraction of relevant documents around each bin of similarity threshold in Figure 5 using the same metric defined in §4.2. We can see that a larger similarity score yields pairs with higher relevance but also more duplicates. Finally, we eliminate near-duplicates using a rule-based filtering approach. The dedup process involves first normalizing text by removing punctuation, converting to lowercase, and eliminating numbers, followed by tokenization using SentencePiece. We then generate "shingles" using 13-token sliding windows within $d_1$. Training pairs are discarded if any shingle from $d_1$ appears in $d_2$.

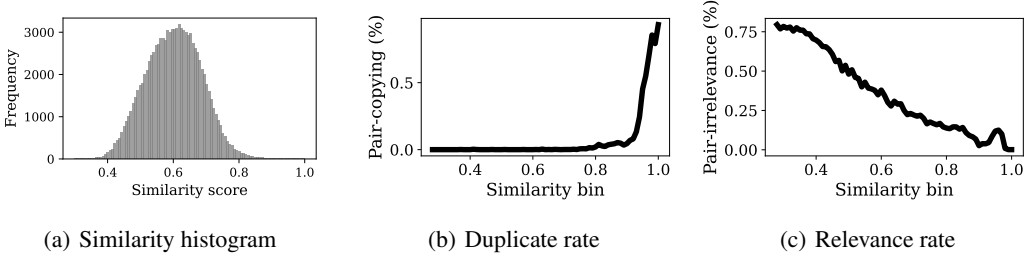

|(a) Similarity histogram | (b) Duplicate rate | (c) Relevance rate |

Figure 5: Analysis of paired data at 200B-scale. Figure 5(a): a histogram of 100K subsampled pairs grouped by their similarity score. Figure 5(b): the fraction of duplicate pairs when we subsample 1K pairs around a specific similarity score. Figure 5(c): same as 5(b) but showing the fraction of relevant documents.

**Synthesizer-tuning** After we collected the cleaned pair data $\mathcal{D}_{\mathrm{ST}}$ (previous step), we perform the synthesizer-tuning with the objective (§3). We initialize the 3B-parameter at the baseline checkpoint and finetune the model with a constant learning rate of 5e-6 and a batch size of 16M tokens per step. Before we settled on this learning rate schedule, we first attempted the cosine decay schedule with a larger learning rate. We found that the generated text has lower quality than our final design with a small, constant learning rate. We measure the Pair-novelty score (defined in §4.2) of the synthesized example for different checkpoints of synthesizer-tuning, and find that longer training results in better Pair-novelty.

**Synthesis at scale**    Finally, we perform the hierarchical sampling procedure defined in §2 with a temperature of 1.0 and top_p threshold 0.9. We apply a rule-based filtering that removes synthesized documents containing repeated occurrences of 13-token shingles. This effectively removes texts with repetition failure. We use vLLM [31] and obtain a throughput of 8.3K tokens per B200 second. This amounts to 2.5K B200 hours for the 200B-scale synthesis, 4.2K B200 hours for the 1T-scale (3B) synthesis, and 8.4K B200 hours for the 1T-scale (6B).

### A.3    Ablation on data mixture ratio

When performing joint training on a mixture of real and synthesized documents for the final SBP run, a natural question arises: how much fraction of synthesized documents to include. In §4, we discussed that we utilized $\|\mathcal{S}_{\text{pretrain}}\|$ =75B for the 200B-scale experiment, $\|\mathcal{S}_{\text{pretrain}}\|$ =125B for the 1T-scale (3B) experiment, and $\|\mathcal{S}_{\text{pretrain}}\|$ =250B for the 1T-scale (6B) experiment. In this section, we present ablation experiments for this design choice.

**200B-scale**    At this smaller scale, we perform a comprehensive sweep over five possible values of $\|\mathcal{S}_{\text{pretrain}}\| \in \{0B, 25B, 50B, 75B, 100B\}$. As seen in Figure 6, different benchmarks exhibit varying behavior when more synthetic data is included during training: the perplexity (OpenWebText2 and LAMBADA) decreases monotonically with increasing synthetic data, while most QA benchmarks display a peak around $\|\mathcal{S}_{\text{pretrain}}\| = 75B$.

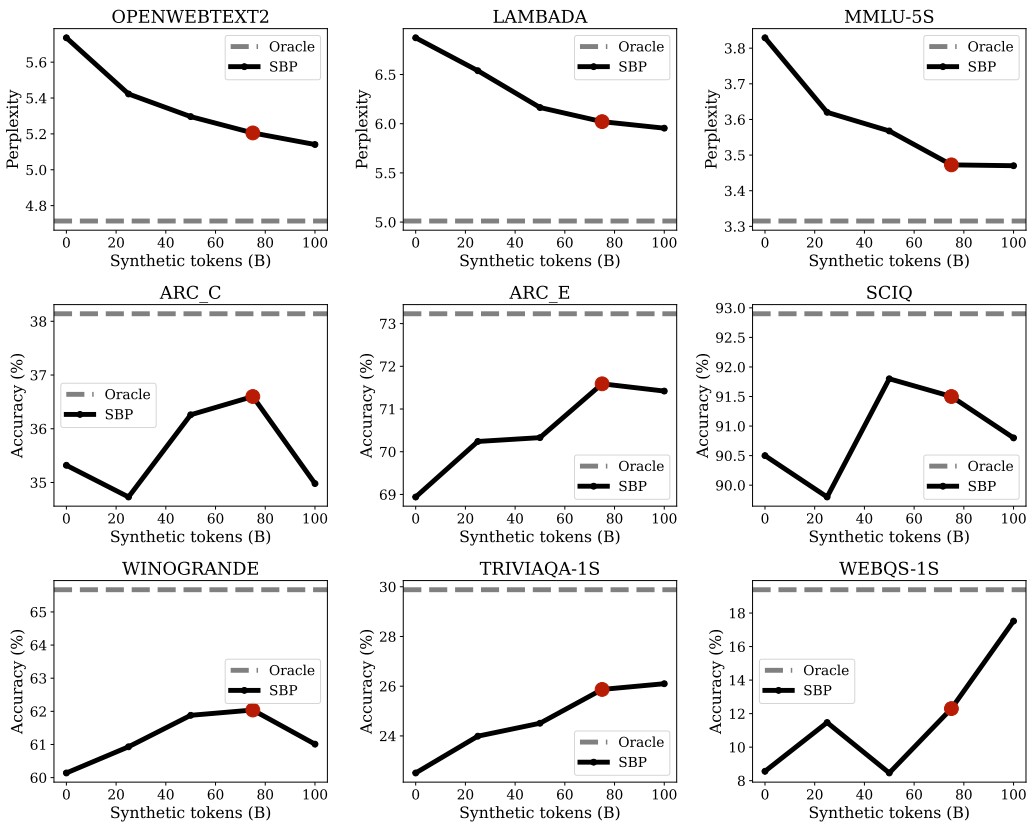

Figure 6: SBP performance with varying synthetic tokens at 200B-scale.

**1T-scale (3B)**    At the 1T-scale, both data synthesis and subsequent joint pretraining become significantly more expensive. Therefore, we evaluate SBP at three different values of the synthetic data $\|\mathcal{S}_{\text{pretrain}}\| \in \{0B, 125B, 250B\}$. As shown in Figure 7, we find that $\|\mathcal{S}_{\text{pretrain}}\|$ =125B produces the best-performing model across all benchmarks except LAMBADA perplexity.

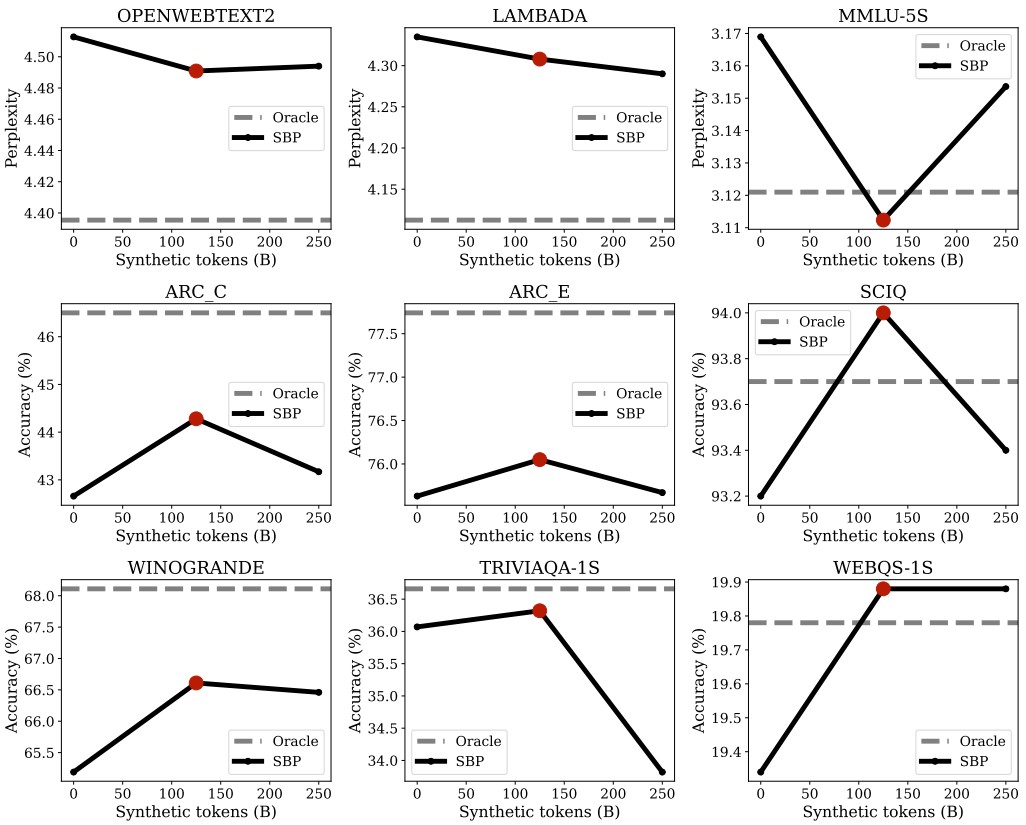

Figure 7: SBP performance with varying synthetic tokens at 1T-scale (3B).

**1T-scale (6B)** We also performed the mixture ratio sweep for the 6B model at the 1T-scale, evaluating $\|\mathcal{S}_{\text{pretrain}}\| \in \{0\text{B}, 125\text{B}, 250\text{B}\}$. As shown in Figure 8, we find that the optimal amount of synthetic data is around 250B, which is higher than the optimal 125B observed for the 3B model.

**Discussion** From this analysis, we can observe a general pattern: the best-performing model is achieved when pretraining is conducted on a mixture of real and synthetic data. Furthermore, the optimal ratio of synthetic data appears to increase with model size (from ~12.5% for 3B to ~25% for 6B). Real internet data has higher quality and therefore merits more repetition. However, as repetition yields diminishing returns, synthetic data could offer another source of signal that real data cannot capture. In contrast, distillation-based research typically finds that training purely on synthetic data yields significantly higher training efficiency. However, this finding is obscured by the fact that such a model eventually converges to the capability of the teacher LM. This contrast reveals that the SBP mechanism does not generate a compressed and denoised representation of knowledge that is more efficient for LM to learn. Instead, it offers an additional source of improvement that real data alone cannot capture.

## A.4 RANDOM PAIRS AND EMBEDDING ANALYSIS

To verify that SBP relies on learning specific inter-document correlations rather than generic data augmentation, we conduct an ablation study where we train the synthesizer on random document pairs. We measure the semantic similarity between the seed document $d_1$ and the target document (either $d_2$ or the synthesized output) using the Qwen3-Embedding-0.6B model.

As shown in Table 3, the paired documents used in SBP exhibit high semantic similarity (0.79), whereas random documents have minimal similarity (0.15). Crucially, the documents generated by the SBP synthesizer maintain a high degree of relevance (0.66) to the seed document. In contrast,

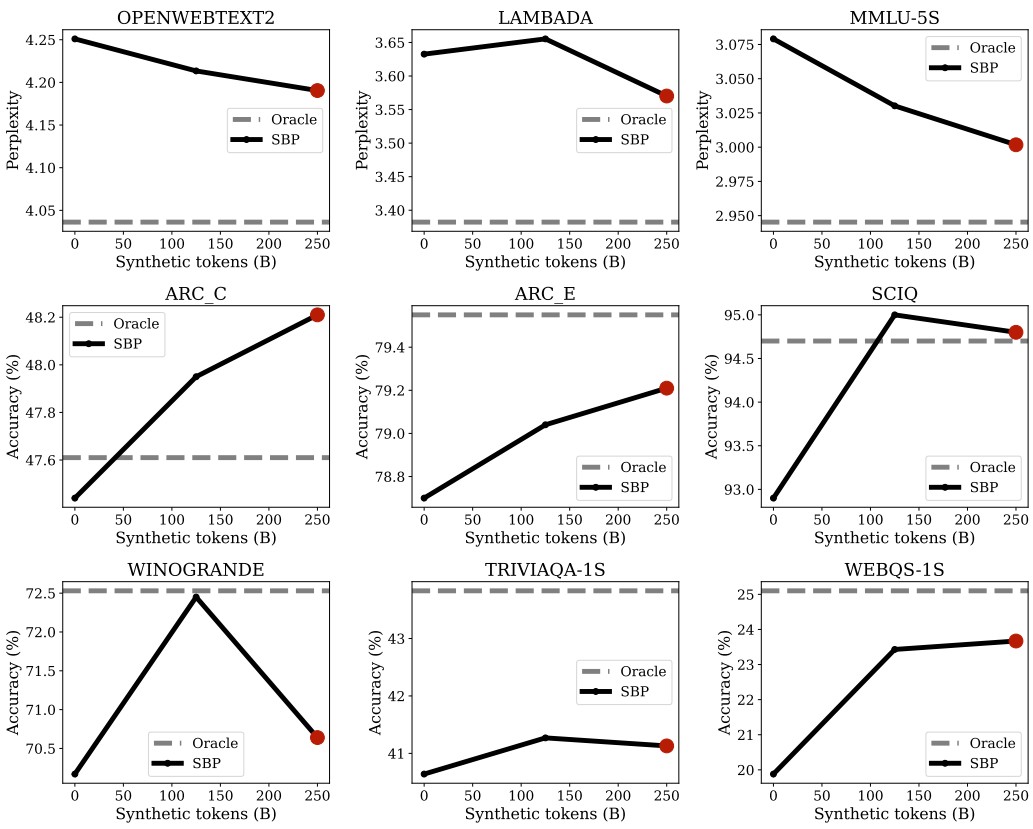

Figure 8: SBP performance with varying synthetic tokens for the 6B model at 1T-scale.

Table 3: Embedding similarity statistics. "Paired documents" refers to the SBP training pairs found by nearest neighbor search. "Random documents" refers to randomly paired documents. "Generated documents (SBP)" refers to the synthetic data generated by the SBP model at 200B-scale (3B). "Generated documents (Random)" refers to the synthetic data generated by the model trained on random pairs. All comparisons are based on the 10B dataset.

| Statistic | Paired docs | Random docs | Generated (SBP) | Generated (Random) |
|-----------|-------------|-------------|-----------------|--------------------|
| Mean | 0.79 | 0.15 | 0.66 | 0.32 |

the model trained on random pairs produces outputs with significantly lower relevance (0.32). This result confirms that the SBP synthesizer successfully learns to preserve semantic relevance from the training pairs, a property that is absent when training on random associations.

# B  ADDITIONAL ANALYSIS OF SYNTHESIZED SAMPLES

## B.1  ANALYZE CONCEPTS IN DOCUMENTS

In this section, we further examine the intermediate hidden mechanisms underlying the document synthesis process. Specifically, we classify the hypothesized concepts inferred from real documents (see Table 4 for details) along two complementary dimensions: concept domains, which denote the broad subject areas or fields a concept belongs to (e.g., science, psychology, health, culture), and concept types, which capture the abstract role or nature of the concept itself (e.g., theory, method, comparison, symbol).

Table 4: Examples of latent concepts $c$ inferred by an external LM (prompts provided in §B.1). From left to right, we provide a summary of the real document, the inferred latent concept, and a summary of the synthesized document.

| Real document summary | Concepts | Synthesized document summary |
|---|---|---|
| Examination of Twitter's impact on journalism | Opportunities arise from Twitter | Guide on Twitter user monetization |
| Family story about kids and doughnuts | Parenting + kids' food catering | Emotional anecdotes of parents treating kids |
| Minor parties' challenges in the U.S. Congress | Minor political parties in the U.S. | Explains U.S. minor parties' history |
| Personal stories/questions about swollen eyes | Causes/treatments of swollen eyes | Non-personal guide to treating swollen eyes. |
| Antarctic carbon fixation mechanisms | How life survives in Antarctic | Antarctic geography and survival adaptations |
| Profile of a belly dancing teacher in the U.K. | Belly dancing as a dance form | General introduction to belly dancing |
| Anxiety about creative work judged in a dream | Dream as personal self-reflection | Description and reflection of personal dreams |
| NYC (yearly/monthly) climate extremes | NYC weather and temperature | QA on NYC July heat and related topics |
| Tutorial for Minecraft block modding | Block editing in Minecraft | Minecraft forum question on removing blocks |
| Cosmic airburst destroys Tall el-Hammam city | Destruction of ancient cities | Tall el-Hammam excavation as a news event |

Table 5: Categorize extracted concepts into domains.

| Concept Domains | Examples |
|---|---|
| Culture (38.74%) | Inter-community conflict in Nigeria, Family-based immigration policy, Reactions to Horrid Henry books, Interracial dating and bias |
| Health (11.89%) | Cosmetic dental appliance, Colistin toxicity in infections, Hair health tips, Portable/home medical diagnostics, Vitamin D and pregnancy outcomes |
| Technology (9.91%) | Recovering deleted phone data, Video editing app review, Flash platform pros and cons, HTML 2.0 draft process, Email attachment processing speed |
| Politics (3.69%) | Iran nuclear negotiations, Student loans policy reform, Democratic primary candidate choice, Catalan independence aftermath |
| Psychology (3.42%) | Differences in personality disorders, Exploring the strange in daily life, Aging and nostalgia, Toxic relationship breakup, Psychology research paper topics |

The distribution of concept domains and types in Table 5 and 6 underscores the multidimensional nature of the knowledge space under consideration. The domains encompass macro-level socio-cultural phenomena, such as Culture, where topics range from inter-community conflict in Nigeria to immigration policy and interracial dating and bias, alongside micro-level issues of individual health and wellbeing, as exemplified in Health. In parallel, the typological classification reveals not only subject matter but also modes of conceptual engagement: Methods comprise formalized procedures (multidimensional poverty measurement, commercial real estate appraisal), Events capture historically situated crises (Mediterranean migrant crisis, BP oil spill nationalization), and Comparisons and Analyses facilitate interpretive framing through juxtapositions (cancer suffering: individual vs. family) and evaluative inquiries (Manchester United player analysis). Collectively, this taxonomy illustrates not only topical diversity but also a spectrum of cognitive orientations.

While real and synthesized documents share the same underlying concept, they differ in multiple ways that merit closer examination. We categorize these differences into a taxonomy of relations using a small ontology. Table 7 illustrates several relationship types, highlighting how synthesized data can reflect multiple facets that vary from real data. These relations range from scope-based distinctions (e.g., specific vs. general), to causal connections (e.g., corruption leading to reform), and to contrastive contrasts (e.g., Constitution articles vs. Articles of Confederation). This diversity demonstrates the rich variation structure that the synthesizer captures and learns.

Table 6: Categorize extracted concepts into abstract types.

| Concept Types | Examples |
| --- | --- |
| Method (9.17%) | Multidimensional poverty measurement, Commercial real estate appraisal, Stop words search duplicates, DAT chemistry exam preparation |
| Event (6.98%) | Mediterranean migrant crisis, BP oil spill nationalization, Paula Abdul stalked, Eminem-Apple music rights lawsuit, Presidents Cup U.S. golf |
| Comparison (5.54%) | Hobbit film adaptation length/cost, Biking as superior transport, Cancer suffering, individual vs. family, Progress critique: 4G vs. alternatives |
| Analysis (5.20%) | Health effects of substances, Thai massage benefits, Scrabble word breakdown, Relationship roles and challenges, Manchester United player analysis |
| Phenomenon (4.95%) | Secret pain; self-destruction, Car-related online humor/pranks, Transnational corporations in globalization, Hippie identity and lifestyle |

Table 7: Categorize relations between real documents $d_1$ and synthesized documents $d_2$.

| Relation Categories | Examples |
| --- | --- |
| Scope relation (8.14%) | $d_1$: Probiotics' possible effects on H1N1 infection
$d_2$: Probiotics' general digestive and immune benefits
Relation: specific application vs general health benefits of probiotics |
| Perspectival relation (5.51%) | $d_1$: Personal, humorous struggles of new bloggers
$d_2$: Objective guide to pros and cons of blogging
Relation: subjective experiences vs objective guidance about blogging |
| Functional relation (4.70%) | $d_1$: Reviews and feedback on "Space Bound" game
$d_2$: Forum troubleshooting for bugs in "Space Bound"
Relation: reviews/feedback vs troubleshooting for the same game |
| Causal relation (2.05%) | $d_1$: DTEK faces corruption probe, financial risk
$d_2$: DTEK nationalized for state-driven energy reform
Relation: corruption/financial issues vs nationalization/energy reform |
| Contrastive relation (1.65%) | $d_1$: Detailed summary of Constitution articles
$d_2$: Overview, flaws of Articles of Confederation
Relation: U.S. Constitution articles vs Articles of Confederation: different foundational documents |

---

**Document Summarize and Concept Analysis Instructions**

In the following, you are given two documents, doc1 and doc2. Doc2 is generated from doc1.

The principle of generation is to first abstract a concept from doc1, and then starting from this concept, generate doc2. Can you guess what this concept is and how doc2 was generated?

Please keep the summary and concepts to be LESS OR EQUAL TO 10 WORDS and format your answer as follows. Highlight the difference between doc2 and doc1 in your doc2_summary:

```
<doc1_summary> summary of doc1 </doc1_summary>
<concept_c> abstract concept from doc1 </concept_c>
<doc2_summary> summary of doc2 built on doc1 given the concept </doc2_summary>
```

Example 1:

```
<doc1_summary> recommendation of local coffee shops in San Diego </doc1_summary>
<concept_c> coffee + San Diego </concept_c>
<doc2_summary> comparison of coffee culture in SD and NYC </doc2_summary>
```

Example 2:

```
<doc1_summary> Patient with swollen eye discusses pain causes & symptoms and seeks for
advice </doc1_summary>
<concept_c> medical symptom of swollen eye </concept_c>
<doc2_summary> A wiki-style article introducing causes and cures for swollen eye
</doc2_summary>
```

Now, give your answer for the following documents:

```
<doc1>
{real_document}
</doc1>

<doc2>
{synthesized_document}
</doc2>
```

## B.2 FACTUALITY ANALYSIS

Table 8: Estimation of the ratio of non-factual documents. We can see that the occurrence factuality error decays as the SP scales up.

|  | Factuality undefined | No factual error | Factual error |
|---|---|---|---|
| **Real data** | 31.44% | 66.74% | 1.81% |
| **Synthetic data (200B-scale)** | 34.43% | 50.47% | 15.09% |
| **Synthetic data (1T-scale)** | 31.91% | 59.43% | 8.65% |

All LM generated synthetic data face the limitation of potentially generating non-factual content due to their probabilistic modeling nature. Moreover, because the internet inherently contains factual inaccuracies, LMs are likely to absorb these errors unless the data is carefully cleaned. During post-training, factuality must also be recalibrated alongside other objectives such as data safety.

SBP relies solely on document-level correlations and does not incorporate human intervention to filter non-factual content. As a result, the generated outputs are also expected to contain factual errors. Interestingly, we observe that the frequency of such errors correlates with the amount of data used in the SBP pipeline. We define a document as having **undefined factuality** if it is primarily subjective or opinion-driven, or if it concerns personal, obscure, or unverifiable entities. In all other cases, the document's factuality is considered **well-defined** and verifiable.

In Table 8, we analyze both the real data and the synthesized data used in the main experiment presented in Section 4.1. Specifically, we consider two types of synthetic datasets: a smaller-scale set initialized with 10B seed tokens, and a larger-scale set initialized with 50B seed tokens. From each source: real data, smaller-scale synthetic data, and larger-scale synthetic data, we randomly sample 10k documents. Each document is then categorized into three bins: **factuality undefined**, **no factual error**, and **factual error**, using LM-as-a-judge. Our analysis shows that synthetic data contains more factual errors than real data. However, as the amount of seed data increases, the factuality of synthetic data improves significantly, approaching that of real data. This finding is consistent with our mideval results in Table 2, where greater seed data availability enables the LM

Table 9: Factuality undefined synthetic text.

| | |
|---|---|
| **Synthetic Text** | Sunday, December 28, 2008 |
| | Tante Leni |
| | Tante Leni is not only my Aunt Leni; she is my Eternity. When my Aunt Leni died a few years ago, she was deeply saddened and I was devastated. She was not the first family member to die, but she was the first I felt so strongly about. Tante Leni was all my parents really had to show for the 25 years they worked as public school teachers and she was the one who had been with them the longest. There was a special place in her heart for my parents. In addition to all that, she was the kind of person who always had a smile and a funny story to share. She was kind and funny and generous. The story that always comes to mind when I think about her is the time she was working at the bank and someone dropped something from the top floor. It was a very large parcel and the workers on the ground floor didn't have the tools to open it. She jumped down to see what was in it. A very large package of champagne appeared and she began gulping it down. Tante Leni and my mother in a portrait they took when my mom was 20. Tante Leni and my parents in a family portrait she took for my mom at 22. Tante Leni and my dad at home when he was working as a dance instructor. When my mom died, she had all the people who had known her since she was a child living in the house. Tante Leni was the oldest, but she was also the best at cleaning, cooking and taking care of the house. When my mom passed away, she went to a rehab center and Tante Leni stayed in the house. |

to capture more factual knowledge and the synthesizer tuning to generate more relevant documents, thereby reducing hallucinations and producing more realistic outputs.

We extend our analysis of factuality errors in synthesized data in Table 10, highlighting the inaccuracies present in the synthetic texts. These include false transfer and timeline claims in football, as well as incorrect institutional, company location, and certification details in the ecolabel example. This underscores the importance of rigorous fact-checking, particularly in areas such as historical events (e.g., sports) and certification standards (e.g., eco-labels).

---

**Factuality detection instructions**

You are a helpful AI assistant. Your task is to evaluate whether the given document has well-defined factuality.

Definitions:

Not well-defined factuality: The document is primarily subjective or opinion-based (e.g., express disapproval of a politician in social media). The document discusses personal, unknown, or unverifiable entities (e.g., a private diary).

Well-defined factuality: The document refers to well-known, identifiable entities (e.g., famous people, historical events, popular movies). Its factual claims can be checked or verified.

Output format:

If the document's factuality is not well-defined, output:

`<not well defined></not well defined>`

If the document's factuality is well-defined and factual, output:

`<well defined>True</well defined>`

If the document's factuality is well-defined but non-factual, output:

`<well defined>False</well defined>`

Now, analyze the following document and provide your answer:

`{document}`

## B.3  Mideval prompts

Before each large-scale synthesis run (on the order of billions of tokens), we begin by synthesizing a small subset of data to evaluate its overall quality, a step we refer to as "mideval". The goal is to maximize the Pair-relevance of the generated data while monitoring Pair-novelty and Non-repetition rates. Although near-duplicates may not directly degrade quality, they reduce the data's overall utility, so we aim to minimize their occurrence. While self-repetition can be removed via rule-based filtering, we still track it as an indicator of the synthesizer's quality. The quality of both the paired training data for synthesizer-tuning and the synthesized document influences the performance of the final model.

We have cited mideval results in many sections throughout the paper. In this section, we present the prompt that was used for Pair-novelty, Pair-relevance, and Non-repetition.

---

**Pair-relevance detection**

You are a helpful AI assistant helping the user to determine if two provided texts are relevant to each other.
The user will provide you two texts in the following format:

```
## Text 1
{text1}

## Text 2
{text2}
```

Your job is to determine if the two texts are relevant enough to be considered as a pair. Relevance means that the two texts are about the same topic, event, entity, person, place, or thing. If two texts talks about completely unrelated topics, they are not relevant.
Please explain your reasoning in your response, and conclude the response in a new line with either "Yes" or "No". Do not end with any other text including punctuation.

---

**Pair-novelty detection**

You are a helpful AI assistant helping the user to determine if two provided texts are near duplicates.
The user will provide you two texts in the following format:

```
## Text 1
{text1}

## Text 2
{text2}
```

Your job is to determine if the two texts are near duplicates, which means they are almost identical, except for some extra white spaces, line breaks, or punctuation. Two texts are not near duplicates if they talk about the same topic but use different language, words, or style.
Please explain your reasoning in your response, and conclude the response in a new line with either "Yes" or "No". Do not end with any other text including punctuation.

---

---

**Non-repetition detection**

You are a helpful AI assistant helping the user to determine if the provided text has repetition issues.

The user will provide you a text in the following format:

```
## Text
{text}
```

Your job is to determine if the text has repetition issues, which means some particular sentence or pattern are repeated more than three times. Some examples of problematic text:

```
## Example 1 of problematic text
I have a list of users in a SharePoint 2010 site.
I want to send an email to all of them.
I have tried the following code:
var email = new MailMessage();
email.From = new MailAddress("");
email.To = new MailAddress("");
email.Subject = "Test";
email.Body = "Test";
var smtp = new SmtpClient();
smtp.Host = "";
smtp.Port = 25;
smtp.Credentials = new NetworkCredential("", "");
smtp.Send(email);
I get the following error:
The server could not send the message.
The server could not send the message.
The server could not send the message.
...

## Example 2 of problematic text
 my Profile
Product Reviews - Send Message
You are responding to the following review:
Submitted: 02-11-2006 by mikeschmid
I have been paddling for 10 years and have owned 10 kayaks.
I have been paddling in the ocean for 5 years and have owned 3 kayaks.
I have been paddling in the ocean in the Pacific Northwest for 3 years and have owned
2 kayaks.
I have been paddling in the ocean in the Caribbean for 2 years and have owned 1 kayak.
I have been paddling in the ocean in the Mediterranean for 1 year and have owned 1 kayak.
I have been paddling in the ocean in the South Pacific for 1 month and have owned 1 kayak.
I have been paddling in the ocean in the South Atlantic for 1 month and have owned
1 kayak.
I have been paddling in the ocean in the Indian Ocean for 1 month and have owned 1 kayak.
I have been paddling in the ocean in the Arctic Ocean for 1 month and have owned 1 kayak.
I have been paddling in the ocean in the Antarctic Ocean for 1 month and have owned
1 kayak ...
```

Please explain your reasoning in your response, and conclude the response in a new line with either "Yes" (which means the text has repetition issues) or "No" (which means the text does not have repetition issues). Do not end with any other text including punctuation.

---

Table 10: Factuality errors detected in synthetic text.

| | |
|---|---|
| **Synthetic Text** | So just how much has Chelsea been prepared to pay for the 34-year-old midfielder? Realistically, the clubs involved should be keeping in the region of £25 million ($38.8 million) and around £30 million ($45.5 million) being bandied about for the player in Italy. With the Blues expected to complete the sale of Cesc Fabregas to Arsenal this week, Lampard appears the logical replacement in midfield, but his bid to extend his contract has hit a roadblock with Chelsea's owners Roman Abramovich and the club being unable to agree to an increase in salary. |
| | Lampard, who played in the Champions League final in Lisbon for Chelsea in 2007, has been linked with a move away from Stamford Bridge this summer, after having his contract with the club indefinitely extended in 2010. There were rumours that he was to leave on a free transfer in the summer of 2011, but it appears that those are again being placed to the side. |
| | Miguel Delaney, from Sky Sports, believes that Lampard could be the highest paid player in the Premier League, and puts the bid that Chelsea are offering him in the region of £6 million ($8.9 million) per year. If Lampard is to join a team in Italy, it seems that this sum would be realistic, but the Premier League is a different beast, and it would be unrealistic for Chelsea to splash out that amount. |
| | As the speculation around his future continues, you can watch the pair discussing the topic in the video below. |
| **Factual Errors** | Cesc Fabregas to Arsenal – Fabregas moved from Arsenal to Barcelona in 2011, then to Chelsea in 2014, not back to Arsenal. |
| | 2007 – The 2014 Champions League final in Lisbon did not involve Chelsea or Lampard; Chelsea won in 2012. |
| | indefinitely extended in 2010 – Lampard did not get an indefinite contract extension in 2010 with Chelsea. |
| **Synthetic Text** | Swanee Glue Brand: First Glue to be Awarded the Swan Ecolabel |
| | Published:27 July 2022 |
| | The global glue stick market is expected to reach USD 3.45 billion by 2028. Adhesives are the first choice of manufacturers in all industries such as food, pharmaceuticals, automotive, aerospace, construction, and packaging. As consumers are increasingly conscious of their carbon footprint and environmental issues, glue manufacturers are aiming to produce products that comply with environmental standards and are effective and cost-effective in their applications. This is why the Swan Ecolabel was established by the Swedish Environment Agency as a certification for sustainable adhesive products. Swanee Glue is one of the world's leading glue brands in glue sticks, and this year its brand received the Swan Ecolabel. UHU is an adhesive brand owned by Bolton Adhesives in the Netherlands, and part of the Italian Bolton Group with a strong agenda for sustainability. |
| | Glue sticks, specifically glue sticks with a wider applicator and swan neck applicators, have the most impact on the environment because they are a consumable item and their impact is greatest when thrown away. Therefore, the Swanee Swan Ecolabel ensures that UHU is part of the solution to the growing demand for sustainable adhesive products. |
| | In order to obtain the Swan Ecolabel, the adhesive must have at least 50% renewable content. Besides this, the glue stick should also contain a higher percentage of recyclable content. UHU meets all these criteria and has a permanent and multi-use applicator. For further information, you can contact UHU receives the Swan Ecolabel |
| **Factual Errors** | Swan Ecolabel was established by the Swedish Environment Agency – The Nordic Swan Ecolabel was established by the Nordic Council of Ministers, not only Sweden. |
| | Netherlands – UHU is based in Germany, not the Netherlands. |
| | 50% renewable content – The Swan Ecolabel requires at least 20% renewable content in adhesives, not 50%. |

## B.4 Synthesized documents from the 1T-scale experiment

In this section, we present additional examples of synthesized documents at the 1T-scale to complement the example given at the 200B-scale in Section 4.2.

| Real document | Synthesis I | Synthesis II |
|---|---|---|
| The Karate Kid

From Wikipedia, the free encyclopedia
(Redirected from Karate Kid)
Jump to: navigation, search
For other uses, see Karate Kid (disambiguation).
The Karate Kid Karate kid.jpg
Directed by John G. Avildsen
Produced by Jerry Weintraub
Written by Robert Mark Kamen
Music by Bill Conti
Cinematography James Crabe
Distributed by Columbia Pictures
Release dates • June 22, 1984 (1984-06-22)
Running time 127 minutes[1]
Country United States
Language English
Budget $8 million[2]
Box office $90.8 million[3]

The Karate Kid is a 1984 American martial arts drama film produced by Jerry Weintraub, directed by John G. Avildsen, written by Robert Mark Kamen, that stars Ralph Macchio, Noriyuki "Pat" Morita and Elisabeth Shue.[4][5] It is an underdog story in the mold of a previous success, Avildsen's 1976 film Rocky ... | The Karate Kid (2010)

The Karate Kid (2010) movie poster
• 115 minutes • Rated PG-13 • Directed by Harald Zwart • Starring: Will Smith, Jackie Chan, RZA, Jaden Smith • More info on IMDB

It's ironic that Daniel LaRusso (Ralph Macchio) and Johnny Lawrence (Jaden Smith) appear in The Karate Kid in their middle ages, because what's most interesting about this remake is the way it updates an old, familiar storyline for a new generation of moviegoers.

Perhaps I shouldn't say "updates" – because that implies that the new version is a significant departure from the 1984 classic. If you're one of those folks who thinks the 1984 version of The Karate Kid is sacrosanct, don't expect anything different here. Just watch both movies and compare the stories and characters.

The story is basically the same as before – young Daniel LaRusso moves to Los Angeles, and soon finds himself beset by Johnny Lawrence and his gang of students from the Cobra Kai karate school, which bullies Daniel whenever it gets the chance ... | The Karate Kid

Trivia: When Daniel and his mom were drinking ice cream and talking about him not talking to them, Daniel turns to his mom and says "We're not just a family. We're a family of good people." After Daniel was sent away, on the front of his underwear (that was on fire), it said "Daniel don't quit on family." The script for this movie originally said that Daniel's mom was in the film but she was cut because the film makers thought she would not look like Elisabeth Shue and they thought the characters could have a less motherly character. Elisabeth Shue actually called the director John Avildsen, and insisted on being in the film. John Avildsen replied that he didn't know whether or not he wanted to add a less motherly character to the film. Shue replied "If you don't, I'm not playing in the movie." The director of the film was also impressed that she did this and he was going to hire her in any case.

Revealing mistake: When Daniel is throwing the bucket at Johnny's head while on the bus, he is supposed to do it several times, yet when he hits Johnny the first time, you can see Johnny fall on the ground ... |
| Real document | Synthesis I | Synthesis II |
| Skip to main content

Are you craving some beach time? If you're escaping down south for spring break, we have some tunes for you. But even if you aren't lucky enough to sprawl in the sand or bask in golden sunshine right now, we'll bring the beach to you! Until your next break, enjoy listening to The Beach Boys and Jack Johnson.

If you're stuck in the cold—or somewhere where the sun is indecisive (I'm looking at you, Midwest)—use this playlist to take you back to sunnier days and remind you that the winter is almost over; soon the sun will last! | Beach Vibe

Any traveller will tell you that if you get on a plane and actually arrive somewhere where the weather is the same for weeks or months, you know you've made a mistake. I've experienced this time and again - it's one of the many things that you just can't fully appreciate or appreciate to the fullest until you feel it in your bones.

My latest trip was to Vietnam last week. I had been planning this trip for a little over a year now and I really hoped it would be everything I had imagined. It was exactly what I'd wanted. The people, the culture, the food, the scenery - it was all perfectly delved into. I'd never been to an Asian country before and so the whole trip was just like discovering a new continent.

It was one of the best places I've ever been ... | Here's a playlist to warm your heart and get you ready to leave for the beach in a few weeks. So many classics from your youth, this playlist packs an emotional punch.

It's kind of a lame playlist. I'm sure a better one exists somewhere on the internet. But on this most northern of days, I like to get a warm chill going and use it as a positive recharge before going outside. This warm chill is a hit with my dogs and they are the best cuddlers in the world, yet when he hits so that makes it perfect for me. This playlist will make your winter that much more bearable and then hopefully you can head to the beach!

You know you want it!

Here's the playlist:

[credit provider="YouTube" url="]

Get our free mobile app |

Figure 9: Comparison of original text with synthesized text variations. On the first row, the real document provides factual information about the 1984 film's production and release. In contrast, the synthesized documents offer subjective commentary, opinions, and behind-the-scenes anecdotes about both the 1984 film and its 2010 remake. On the second row, the synthesized documents are continuations of the real document.

## C Additional pretraining results

### C.1 Two epochs validation

When designing the oracle experiment for 1T-scale, we noted that we use 482B tokens repeated twice as a proxy for training on 1T unique tokens. This is because the DCLM-baseline [34] dataset contains 80% duplicates, which hinders our evaluation. We validate our choice by scaling down the experiment to a 400B scale, where we had sufficiently many unique tokens. As seen in Table 11, 200B tokens repeated twice yield nearly identical performance to 400B unique tokens. This finding is consistent with the observation from [39] where repetition up to 4 times yields nearly no performance degradation.

### C.2 Model Scaling

An alternative approach to leveraging additional compute is to use a larger model. In this section, we examine the benefits of fixing a training token budget, but using a 6B-parameter model (Table 12).

Table 11: Performance comparsion with 200B tokens repeated twice vs. 400B unique tokens for the 3B model. We can see that the two models yield similar performance.

| Benchmark | 2x200B | 1x400B |
|---|---|---|
| *Perplexity on held-out data ↓* | | |
| OpenWebText2 | 4.55 | 4.54 |
| LAMBADA | 4.49 | 4.46 |
| Five-shot MMLU | 3.19 | 3.17 |
| *QA accuracy ↑* | | |
| ARC-Challenge (0-shot) | 38.31 | 41.47 |
| ARC-Easy (0-shot) | 73.11 | 75.29 |
| SciQ (0-shot) | 93.80 | 93.30 |
| Winogrande (0-shot) | 64.96 | 63.93 |
| TriviaQA (1-shot) | 32.51 | 34.35 |
| WebQS (1-shot) | 18.75 | 13.58 |
| **Average QA accuracy** | 53.57 | 53.65 |

We conduct a pretraining experiment in a 200B-scale setting, replacing a 3B-parameter model with a 6B-parameter model. In Table 13, we observe that the 6B-parameter model consistently outperforms the baseline method, indicating that it effectively utilizes the additional computational resources available. Comparing SBP with the 6B-parameter model, we see that one performs better on some benchmarks while the other performs better on others. This suggests the benefits offered by SBP are orthogonal to the benefits provided by having a larger model, offering the potential to combine both approaches to obtain an even better model.

Table 12: 6B-parameter model setup.

| Total Params. | 3B | 6B |
|---|---|---|
| $\ell_{\text{context}}$ | 4096 | 4096 |
| $n_{\text{vocab}}$ | 49152 | 49152 |
| $n_{\text{layers}}$ | 26 | 32 |
| $d_{\text{model}}$ | 3072 | 4096 |
| $d_{\text{ffn}}$ | 8064 | 13056 |
| $n_{\text{heads}}$ | 24 | 32 |
| $n_{\text{kv\_heads}}$ | 8 | 8 |

Table 13: 200B-scale experiments with model scaling. The first three columns are identical to Table 1. The last column shows the performance of training a 6B model under a 200B training token budget with 10B unique tokens.

| Benchmark | Baseline | SBP | Oracle | 6B-model |
|---|---|---|---|---|
| *Perplexity on held-out data ↓* | | | | |
| OpenWebText2 | 5.74 | -0.53 | -1.02 | -0.36 |
| LAMBADA | 6.87 | -0.85 | -1.86 | -1.10 |
| Five-shot MMLU | 3.83 | -0.36 | -0.51 | -0.13 |
| *QA accuracy ↑* | | | | |
| ARC-Challenge (0-shot) | 35.32 | +1.28 | +2.82 | +3.42 |
| ARC-Easy (0-shot) | 68.94 | +2.65 | +4.29 | +0.67 |
| SciQ (0-shot) | 90.50 | +1.00 | +2.40 | +0.80 |
| Winogrande (0-shot) | 60.14 | +1.90 | +5.53 | +2.92 |
| TriviaQA (1-shot) | 22.51 | +3.36 | +7.37 | +3.11 |
| WebQS (1-shot) | 8.56 | +3.74 | +10.83 | +5.22 |
| **Average QA accuracy** | **47.66** | **+2.32** | **+5.54** | +2.69 |

