# OpenReview forum: "Synthetic Bootstrapped Pretraining"
_ICLR.cc/2026/Conference — ICLR 2026 Poster_

### Official Review · Reviewer_6oUP · 2025-10-28

**Soundness:** 2
**Presentation:** 3
**Contribution:** 2
**Rating:** 4
**Confidence:** 4

**Summary:**

The paper proposes Synthetic Bootstrapped Pretraining (SBP), a framework that improves language model pretraining by generating synthetic documents from existing data.

SBP first finds semantically related document pairs $(d_1, d_2)$, trains a conditional model to predict $d_2$ from $d_1$, and then uses this model to synthesize a large corpus for joint training.

Experiments with a 3B-parameter model under compute-matched setups (200B and 1T tokens) show consistent improvements over repetition baselines and recover up to ~50% of the oracle gain from 20× more unique data.
Quantitative analysis confirms that SBP-generated data maintains diversity and factuality comparable to real text.

The paper also offers a Bayesian view, interpreting SBP as implicitly learning latent “concept” structures across documents, providing a principled explanation for its data-efficiency benefits.

**Strengths:**

1. SBP introduces a principled way to exploit inter-document relations for self-synthetic pretraining without external teachers.

2. Compute-matched experiments at 200B/1T scales show consistent gains and high-quality synthetic data, recovering about half of the oracle improvement.

**Weaknesses:**

1. While SBP claims to capture inter-document correlations, the experiments mainly report aggregate metrics (loss, accuracy) without direct analysis of cross-document dependencies or representation changes. The authors should add the following experiments:

    - Evaluate whether the SBP-trained model can generate $\hat{d}_2$ from $d_1$ that semantically aligns with the true $d_2$ , proving it learns document-to-document conditional structure.

    - Compare embedding distances of paired vs. random documents, see if SBP pulls semantically related but stylistically different pairs closer than the baseline, which proves structural alignment in representation space.

2. The repetition baseline may not reflect realistic industry data-mixing or quality-weighted sampling strategies, possibly inflating SBP’s relative gains.

3. Add ablation: use random document pairs to train the synthsizer.

4. The paper does not clearly state whether the synthesizer-tuning and data generation costs are included in the compute-matched budget, leaving uncertainty about true efficiency.

**Questions:**

1. The paper argues that SBP captures inter-document structure ignored by standard pretraining.

    - Could the authors provide a more direct analysis or evidence of such cross-document dependencies (e.g., similarity shifts in embedding space, conditional generation tests, or document-pair prediction tasks)?

2. The 200B-scale experiment uses a 37.5% synthetic and 62.5% real data ratio.

    - How sensitive are the results to this ratio?

    - Would a smaller amount of synthetic data (e.g., 10–20%) still bring noticeable gains?

3. The paper presents SBP as compute-matched to repetition baselines, but it remains unclear whether the synthesizer training and data generation costs are included in the total compute budget.

    - Could the authors clarify how these costs were accounted for, and whether SBP remains more efficient when end-to-end compute (including synthesis) is considered?


4. The synthesizer is initialized from a pretrained LM, which may introduce prior knowledge not attributable to SBP itself.

    - Could the authors clarify how much this initialization contributes to the overall improvement?

    - Have they tried training the synthesizer from scratch or using weaker initialization to isolate SBP’s effect?

---

> ### Author Response · Authors · 2025-11-25
>
> > Evaluate whether the SBP-trained model can generate d2_hat from d1 that semantically aligns with the true d2, proving it learns document-to-document conditional structure.
>
> Thanks a lot for the insightful suggestion! We provide semantical alignment analysis between d1 and d2_hat in the following places: (1) In Figure 3, we give an example (more examples in Figure 8) of generated text for a particular d1, and we can see that the two distinct d2_hat are semantically related to d1; (2) In Table 2, we provide a quantitive analysis where we subsample pairs of (d1, d2_hat) and use LM-as-judge to determine if two documents are semantically aligned, reported as the “Pair-irrelevance” ratio column; (3) In Table 3, we show 10 examples of (d1, d2_hat) and describe how they are connected via a common underlying concept. Unfortunately, there is no such notion of “true d2”. The synthesizer-tuning process is a one-to-many mapping (described in Section 2.2). More specifically, one particular d1 may correspond to 200 different d2 during synthesizer tuning. As a result, the synthesizer learns a distribution over the potential d2’s.
>
> > Compare embedding distances of paired vs. random documents, see if SBP pulls semantically related but stylistically different pairs closer than the baseline, which proves structural alignment in representation space. The paper argues that SBP captures inter-document structure ignored by standard pretraining. Could the authors provide a more direct analysis or evidence of such cross-document dependencies (e.g., similarity shifts in embedding space, conditional generation tests, or document-pair prediction tasks)?
>
> This is a great suggestion! We perform the suggested experiment below:
> | Statistic | Paired documents|   Random documents| Generated documents|
> | :---      |            ---: |              ---: |               ---: |
> | Mean      |          0.7892 |            0.1482 |             0.6610 |
> | Std       |          0.0328 |            0.0703 |             0.1480 |
> | 25%       |          0.7626 |            0.0984 |             0.5689 |
> | 50%       |          0.7804 |            0.1478 |             0.6829 |
> | 75%       |          0.8074 |            0.1897 |             0.7643 |
>
> The first column, “Paired documents”, is the documents we selected using nearest neighbor search and used for synthesizer tuning. The second column, “Random documents”, means pairs of random documents as you suggested. The third column is the generated documents that are used in SBP training. We can see that the “Paired documents” column has very high relevance (0.78), whereas the “Random documents” column has minimal relevance (0.14). The “Generated documents” shows substantially higher relevance (0.66) compared with random documents, preserving the relevance it learned from the “Paired document” column. This proves that SBP indeed pulls semantically related but stylistically different pairs closer than the baseline.
>
> > The repetition baseline may not reflect realistic industry data-mixing or quality-weighted sampling strategies, possibly inflating SBP’s relative gains.
>
> Thanks a lot for the great comment. We have discussed this in Appendix A.1 that we simplified the mixture nature of the real pretraining corpus. We will move this to the early part of the main body in the camera-ready version of the paper.
>
> > Add ablation: use random document pairs to train the synthsizer.
>
> This is a great suggestion. We performed synthesizer-tuning on a corpus of random pairs of documents and used it to generate synthetic data. Below, we report the semantic similarity measured by the embedding model between the seed document and the generated document:
>
> | Statistic | SBP synthesizer-tuning | Random synthesizer-tuning |
> | :---      |                   ---: |                      ---: |
> | Mean      |                 0.6610 |                    0.3159 |
> | Std       |                 0.1480 |                    0.1391 |
> | 25%       |                 0.5689 |                    0.2110 |
> | 50%       |                 0.6829 |                    0.2907 |
> | 75%       |                 0.7643 |                    0.3991 |
> We can see that we are performing synthesizer-tuning using pairs of randomly selected documents; the synthesized document pertains to substantially lower relevance to the original document. This ablation demonstrates the effectiveness of training on pairs of related documents.

---

> > ### Author Response · Authors · 2025-11-25
> >
> > > The paper does not clearly state whether the synthesizer-tuning and data generation costs are included in the compute-matched budget, leaving uncertainty about true efficiency. The paper presents SBP as compute-matched to repetition baselines, but it remains unclear whether the synthesizer training and data generation costs are included in the total compute budget. Could the authors clarify how these costs were accounted for, and whether SBP remains more efficient when end-to-end compute (including synthesis) is considered?
> >
> > Thanks a lot for the feedback! The compute-matched budget does not include the cost of synthesizer-tuning and data-generation. We stated this at the beginning of Section 3.1: “Operationally, we control the training compute by controlling the total tokens seen during training, which is proportional to the training FLOPs given a fixed batch size and context window.” Indeed, we only control the final training compute across three runs. SBP focuses on the most cutting-edge “scaling wall” scenario, where we run out of the finite internet data and would like to spend more compute to get better performance in this data-constrained setting.
> >
> > > The 200B-scale experiment uses a 37.5% synthetic and 62.5% real data ratio. How sensitive are the results to this ratio? Would a smaller amount of synthetic data (e.g., 10–20%) still bring noticeable gains?
> >
> > This is a great question. We performed this ablation in Appendix A.3 of the paper. As we can see in Figure 6, smaller ratios (e.g., 10-20%) of synthetic data yield smaller gains.
> >
> > > The synthesizer is initialized from a pretrained LM, which may introduce prior knowledge not attributable to SBP itself. Could the authors clarify how much this initialization contributes to the overall improvement? Have they tried training the synthesizer from scratch or using weaker initialization to isolate SBP’s effect?
> >
> > To clarify the setup, the pretrained LM is also trained on the seed data itself (as opposed to a third-party base model such as DeepSeek-V3).  In fact, the pretrained LM is the “baseline” column in Table 1. This does not violate SBP’s “no distillation” assumption, as SBP and baseline have access to the same amount of unique documents. As a result, all improvements stem from the SBP algorithm itself, and the synthesizer is indeed trained from scratch – first go through the normal pretraining as baseline did, and then go through the synthesizer-tuning.

---

### Official Review · Reviewer_9CYh · 2025-10-29

**Soundness:** 3
**Presentation:** 3
**Contribution:** 2
**Rating:** 4
**Confidence:** 3

**Summary:**

This paper introduces Synthetic Bootstrapped Pretraining (SBP), a new large-scale pretraining procedure that aims to capture inter-document correlations that standard LM pretraining overlooks. SBP identifies semantically similar document pairs, trains a conditional “synthesizer” model to generate one document from another, and then uses it to synthesize new text for joint training. The method is evaluated using compute-matched experiments on a 3B model trained up to 1T tokens. Results show consistent improvement over repetition-based baselines, recovering about 40–50% of the performance gains achievable by an oracle model with access to 20× more unique data. The authors also present a Bayesian interpretation of SBP and provide qualitative and quantitative analyses of the generated data.

**Strengths:**

1. Introduces a new dimension of pretraining that models inter-document relations rather than intra-document token dependencies.
2. Clear structure and logical flow from motivation.

**Weaknesses:**

1. Can a 1.5B model trained on a 200B dataset generate data that is both relevant to the source documents and diverse?
2. What is the computational resource consumption across the three stages?
3. The related evaluation metrics are highly dependent on the LLM used for judgment. What is the judged LLM?
4. The "pair copying" metric should not be compared solely with the source documents but rather with the entire training dataset. This is because the model might output documents that are similar to others in the dataset, which would contribute little to overall improvement.
5. Comparing Figure 6 and Figure 7 reveals that as the dataset size increases, the proportion of synthetic data required decreases, which to some extent also illustrates this issue.

**Questions:**

See Weaknesses.

---

> ### Author Response · Authors · 2025-11-25
>
> > Can a 1.5B model trained on a 200B dataset generate data that is both relevant to the source documents and diverse?
>
> This is a great question! We briefly experimented with the 1B model and found it can’t generate coherent text when trained on only 200B (10B unique tokens repeated 20 times) tokens from DCLM. We speculate the 1.5B model probably wouldn’t work very well. As we discussed in Section 6.1, synthetic data is a technique that is difficult to scale down. The performance improvement emerges at a large scale.
>
> > What is the computational resource consumption across the three stages?
>
> We presented the compute resource for each stage in the paper when the stage was introduced. As a quick summary, for the 1T-scale experiment, the nearest-neighbor search over 60M documents costs 155M CPU hours. The subsequent synthesis costs 4.2K B200 hours. The final pretraining costs 59K v5p-TPU hours.
>
> > The related evaluation metrics are highly dependent on the LLM used for judgment. What is the judged LLM?
>
> We used `gpt-4.1-2025-04-14`.
>
> > The "pair copying" metric should not be compared solely with the source documents but rather with the entire training dataset. This is because the model might output documents that are similar to others in the dataset, which would contribute little to overall improvement.
>
> Thanks a lot for this insightful question. Our report on pair-copying is intended to shed light on the data synthesizer's behavior, not to argue that a lower pair-copying rate contributes to the final performance improvement. For the specific experiment suggested, it’s challenging to iterate over the entire real dataset for each (or a reasonable subset) of synthetic documents to detect pair-copying. This is because there are 60 M pretraining documents, and iterating over them for each synthetic data point is infeasible. For the nearest-neighbor search, this is possible thanks to the ScaNN index tree (https://research.google/blog/announcing-scann-efficient-vector-similarity-search/) for a single shard of documents. As a result, we only need to iterate over the shards that are built in parallel. There is no such clean implementation for pair-copying detection. Moreover, the final pretraining mix comprises both real and synthetic data. At the 200B-scale, the real data is repeated 12.5 times, so the repetition is already there.
>
> > Comparing Figure 6 and Figure 7 reveals that as the dataset size increases, the proportion of synthetic data required decreases, which to some extent also illustrates this issue.
>
> Thanks a lot for the great comment! The difference between Figures 6 and 7 means that, holding the model size constant, having more unique real data means we should apply less synthetic data. This is because the large volume of real data has already saturated the capacity of a small model, so adding large amounts of synthetic data yields limited benefits. We believe this is a separate issue from the “pair-copying” discussion above.
>
> To further test this hypothesis, we conduct an additional 1T-scale experiment using a larger 6B-parameter model.
>
> | Benchmark                  | Baseline |    SP | Oracle |
> | :------------------------- | -------: | ----: | -----: |
> | **Perplexity (lower ↓)** |          |       |        |
> | OpenWebText2               |     4.25 | -0.06 |  -0.21 |
> | LAMBADA                    |     3.63 | -0.06 |  -0.25 |
> | MMLU (5-shot)              |     3.08 | -0.08 |  -0.13 |
> | **QA Accuracy (higher ↑)** |          |       |        |
> | ARC-Challenge (0-shot)     |    47.44 | +0.77 |  +0.17 |
> | ARC-Easy (0-shot)          |    78.70 | +0.51 |  +0.85 |
> | SciQ (0-shot)              |    92.90 | +1.90 |  +1.80 |
> | Winogrande (0-shot)        |    70.17 | +0.47 |  +2.36 |
> | TriviaQA (1-shot)          |    40.64 | +0.49 |  +3.19 |
> | WebQS (1-shot)             |    19.88 | +3.79 |  +5.22 |
> | **Average** |**58.29** |**+1.32**|**+2.26**|
>
> We sweep the amount of synthetic data at {125B, 250B} (same choice as the experiment with 3B-model in Figure 7 you mentioned) and find that the best performing model is obtained at 250B. This suggests that with larger model, pretraining would benefit from more synthetic data, supporting the perspective that the optimal synthetic data ratio scales positively with model size. On a side note, the relative improvement of the 6B model is 1.32/2.26~60% compared with ~40% for the 3B model. This means SBP delivers more improvement with the baseline with a larger model.

---

> > ### Comment · Reviewer_9CYh · 2025-11-26
> >
> > Thank you for your detailed response. As the authors stated, “We speculate the 1.5B model probably wouldn’t work very well.” I still have some concerns about the completeness of the paper’s conclusions. With a larger model, it might be necessary to conduct fair comparative experiments on a larger amount of data. For now, I will maintain my score.

---

> > > ### Author Response · Authors · 2025-11-26
> > >
> > > Thanks a lot for the continued response! We would appreciate clarification on "fair comparative experiments on a larger amount of data," as we have already conducted evaluations across three distinct scales: a 3B model trained on 200B tokens, a 3B model trained on 1T tokens, and a 6B model trained on 1T tokens. These experiments reveal a consistent and positive scaling trend, where the relative performance gain of SBP actually increases with model size, recovering 60% of the Oracle gap for the 6B model compared to 40% for the 3B model.
> > >
> > > We also wish to emphasize that our 1T-scale experiment with the 6B model already operates at a massive compute regime, utilizing approximately 1e22 FLOPs, which corresponds to roughly 50% of the total compute budget used to train the Llama 2 7B base model (2T total training tokens as opposed to 1T, https://arxiv.org/pdf/2307.09288).
> > >
> > > Finally, our study is explicitly framed within the "data-constrained regime" where high-quality unique data is the limiting factor; therefore, comparing against settings with unlimited or significantly larger amounts of unique data would deviate from the core problem statement of addressing the impending data wall.

---

### Official Review · Reviewer_s1yS · 2025-10-30

**Soundness:** 2
**Presentation:** 3
**Contribution:** 3
**Rating:** 4
**Confidence:** 3

**Summary:**

This paper introduces Synthetic Bootstrapped Pretraining (SBP), a novel framework aimed at improving data efficiency by explicitly modeling inter-document correlations within the pretraining dataset. The authors conduct rigorous compute-matched large-scale experiments (3B model, up to 1T tokens) demonstrating that SBP significantly outperforms the traditional repetition baseline and effectively reduces the performance gap to the "Oracle" upper bound (a model with access to more unique data). The concept is novel, the experiments are solid, the results are convincing, and the inclusion of a theoretical explanation based on a Bayesian hierarchical concept model makes this a valuable and highly relevant research direction for addressing the imminent "data scaling wall."

**Strengths:**

1. SBP creatively addresses the issue of exhausting high-quality text data. By learning a conditional generator $P(d_2|d_1)$ from document pairs to synthesize new data, it serves as a powerful complement to traditional token-level autoregressive training, leveraging latent structural signals in the data.

2. The experiments utilize a 3B-parameter model, making the scale relevant for frontier LM development. Crucially, the compute-matched setup ensures a fair comparison across SBP, the repetition baseline, and the Oracle.

3. The results (Table 1 in the original paper) show SBP bridges 42% to 48% of the QA accuracy gap to the Oracle, confirming that the synthesized data $S_{pretrain}$ provides a real and effective signal.

**Weaknesses:**

1. The quantitative analysis (Tables 2 and 7 in the original paper) shows an alarmingly high non-factual error rate of 15.1% for the 200B-scale synthetic data (compared to 1.81% for real data). Although this rate decreases to 8.7% at the 1T-scale, it remains a significant concern. By the way, the caption in Table 2, `are not are' should be `are not'?

2. The paper acknowledges in Section 6 that the initial nearest neighbor search relies on the external Qwen3-0.6B Embedding model. Why choose such a model, while efficiency is a valid reason, this limitation detracts from the "Bootstrapped Pretraining" claim.

3. The $d_1 \to c \to d_2$ relationship captured by SBP is fundamentally a statistical dependency (or association) based on a shared latent concept $c$. While this process involves an "implicit posterior inference," framing this learned relationship as "Causality" in a statistical context is inaccurate and can be misleading, as it does not imply a strong intervention relationship.

**Questions:**

See weakness.

---

> ### Author Response · Authors · 2025-11-25
>
> > The quantitative analysis (Tables 2 and 7 in the original paper) shows an alarmingly high non-factual error rate of 15.1% for the 200B-scale synthetic data (compared to 1.81% for real data). Although this rate decreases to 8.7% at the 1T-scale, it remains a significant concern. By the way, the caption in Table 2, “are not are” should be “are not'?
>
> Thanks a lot for this insightful observation. Indeed, we observed a favorable scaling trend with larger pretraining datasets, but the non-factual rate remains high. To further solidify our scaling claim, we repeat SBP at the 1T-scale with a 6B-parameter model as opposed to the 3B-paramter as reported in the paper. This scaling imposes significant infrastructure challenges, but we are pleased to see that the non-factual rate further declined to **6.54%**. This shows that the non-factual rate decreases as we use more data or larger models. On the other hand, the base-level non-factual rate for real data is fixed. In practice, we would perform pretraining up to 30T unique tokens with 3T (100B activated) parameters. This is significantly larger than the largest experiment shown here, which uses a 1T-token dataset and a 6B-parameter dense model. As a result, this problem should be alleviated by applying our method to a real language model. We defer this to future work due to the resource limit during the rebuttal period.
>
> Additionally, the new 6B model delivers 60% relative improvement as opposed to 40% improvement for 3B model (the main result reported in the paper). This means not only SBP scales, it also delivers stronger results with a larger model.
>
> | Benchmark                  | Baseline | SP | Oracle |
> | :------------------------- | -------: | ----: | -----: |
> | **Perplexity (lower ↓)** |          |       |        |
> | OpenWebText2               |     4.25 | -0.06 |  -0.21 |
> | LAMBADA                    |     3.63 | -0.06 |  -0.25 |
> | MMLU (5-shot)              |     3.08 | -0.08 |  -0.13 |
> | **QA Accuracy (higher ↑)** |          |       |        |
> | ARC-Challenge (0-shot)     |    47.44 | +0.77 |  +0.17 |
> | ARC-Easy (0-shot)          |    78.70 | +0.51 |  +0.85 |
> | SciQ (0-shot)              |    92.90 | +1.90 |  +1.80 |
> | Winogrande (0-shot)        |    70.17 | +0.47 |  +2.36 |
> | TriviaQA (1-shot)          |    40.64 | +0.49 |  +3.19 |
> | WebQS (1-shot)             |    19.88 | +3.79 |  +5.22 |
> | **Average** |**58.29** |**+1.32**|**+2.26**|
>
> > The paper acknowledges in Section 6 that the initial nearest neighbor search relies on the external Qwen3-0.6B Embedding model. Why choose such a model, while efficiency is a valid reason, this limitation detracts from the "Bootstrapped Pretraining" claim.
>
> Thanks a lot for the sharp comment! Indeed, we discussed this in the paper's limitations section. We believe the same result can be obtained using the hidden activation of the pretrained transformer model (the baseline in Table 1), which doesn’t use any new data. In fact, the kNN-LM (https://arxiv.org/abs/1911.00172) and ResMem (https://arxiv.org/abs/2302.01576) adopt this embedding technique. However, our infrastructure constraints pose significant challenges to repurposing our pretrained LM as an embedding model compatible with vLLM. We therefore resort to the Qwen3-0.6B-Embedding. We intentionally chose the 0.6B model as it’s much smaller than our 3B model in the paper.
>
> > The d1->c->d2 relationship captured by SBP is fundamentally a statistical dependency (or association) based on a shared latent concept. While this process involves an "implicit posterior inference," framing this learned relationship as "Causality" in a statistical context is inaccurate and can be misleading, as it does not imply a strong intervention relationship.
>
> We appreciate the great comment! However, we never claimed the learned relationship is a “Causality” in any manner. As noted in your comment, we believe there is no such intervention. Please let us know if we have misinterpreted your comment. Sorry in advance!

---

### Official Review · Reviewer_Vkk4 · 2025-11-01

**Soundness:** 3
**Presentation:** 3
**Contribution:** 3
**Rating:** 6
**Confidence:** 3

**Summary:**

This paper proposes a method to generate synthetic data for pre-training in constrained data settings. The method relies on a document generation model that learns to generate new diverse documents from given ones. Concretely, an LM is pre-trained on a dataset D_pretrain (standard next token prediction) and then fine-tuned on the document generation task. To fine-tune the document generator, pairs of input-output documents are created by pairing related documents in D_pretrain. The document generator is then used to generate a synthetic corpus of documents D_synthetic. An LM is then pre-trained on D_pretrain+D_synthetic. The authors also discuss a Bayesian interpretation of the standard data pre-training objective versus the document generator (synthesizer)fine-tuning one; the former marginalizes out latent concepts from which documents are generated while the second models the posterior over these concepts.

**Strengths:**

- An intelligent idea to augment with synthetic data that carries new signals (compared to the repetition baseline).

**Weaknesses:**

- There are known issues related to biases set up in pre-training (e.g., [1]). The authors should probably report analysis with respect to this. In particular because of the document generator.

[1] https://openreview.net/pdf?id=KQhUEoPmJy

- From the experiments it is not clear how the improvements brought by the data synthesizer would generalize to larger models. From Table 2 it seems to indicate that with scale, the generated documents might be closer to the input documents. Still the idea is interesting and could be useful for the community.

**Questions:**

- Line 135 DATA-CONTRAINED > CONSTRAINED

- For the Quantitative analysis in Section 4.2. It would make sense to add metrics based on novel n-grams, e.g.,  percentage of new n-grams appearing in the generated documents or those used in [2], to better reflect the deviation of the generated documents from the seed ones.

[2] https://aclanthology.org/N18-1065/

- Statistics about the length of synthetically generated documents?

---

> ### Author Response · Authors · 2025-11-25
>
> > There are known issues related to biases set up in pre-training (e.g., [1]). The authors should probably report analysis with respect to this. In particular because of the document generator.
>
> Thanks a lot for the reference! We will discuss it in the final version of the paper: The biases planted in the model are more a property of the dataset SBP algorithm used than of the algorithm itself. In our case, we performed qualitative evaluations of the synthesized data, including their topic (Table 3), factuality (Table 9), and their relations to the seed document (Table 6) in the appendix.
>
> > From the experiments it is not clear how the improvements brought by the data synthesizer would generalize to larger models. From Table 2 it seems to indicate that with scale, the generated documents might be closer to the input documents. Still the idea is interesting and could be useful for the community.
>
> Thanks a lot for this insightful question! To demonstrate that the improvements brought by the data synthesizer indeed generalize to larger models, we repeat our 1T-scale experiment but with a 6B-parameter model instead of the 3B-parameter model presented in the paper.
>
> | Benchmark                  | Baseline |    SP | Oracle |
> | :------------------------- | -------: | ----: | -----: |
> | **Perplexity (lower ↓)** |          |       |        |
> | OpenWebText2               |     4.25 | -0.06 |  -0.21 |
> | LAMBADA                    |     3.63 | -0.06 |  -0.25 |
> | MMLU (5-shot)              |     3.08 | -0.08 |  -0.13 |
> | **QA Accuracy (higher ↑)** |          |       |        |
> | ARC-Challenge (0-shot)     |    47.44 | +0.77 |  +0.17 |
> | ARC-Easy (0-shot)          |    78.70 | +0.51 |  +0.85 |
> | SciQ (0-shot)              |    92.90 | +1.90 |  +1.80 |
> | Winogrande (0-shot)        |    70.17 | +0.47 |  +2.36 |
> | TriviaQA (1-shot)          |    40.64 | +0.49 |  +3.19 |
> | WebQS (1-shot)             |    19.88 | +3.79 |  +5.22 |
> | **Average** |**58.29** |**+1.32**|**+2.26**|
>
> This table follows the same evaluation protocol as the main results Table 1 in the paper.  We can see that the relative improvements jump to 1.32/2.26 ~ 60% (for 3B models, this number was only 40%). This is a even larger margin of improvement than the 3B-parameter models. This experiment indicates that the SBP improvement generalizes to larger models, and the improvement is even larger compared with a small model.
>
> > For the Quantitative analysis in Section 4.2. It would make sense to add metrics based on novel n-grams, e.g., percentage of new n-grams appearing in the generated documents or those used in [2], to better reflect the deviation of the generated documents from the seed ones.
>
> Thanks a lot for this insightful suggestion. We perform 16-gram novelty detection and report the result below. Everything except the last column is copied from Section 4.2 for your reference; only the last column is newly added.
>
> |            | Repetition ↓ | Duplicate@1M ↓ | Non-factual ↓ | Pair-irrelevance ↓ | Pair-copying ↓ | 16-gram-repetition (new column) |
> | :---       |         ---: |           ---: |          ---: |               ---: |           ---: |                            ---: |
> | 200B-scale |         4.3% |           0.8% |         15.1% |              25.6% |           0.1% |                          0.169% |
> | 1T-scale   |         3.9% |           0.8% |          8.7% |               7.8% |           0.9% |                          0.147% |
> | Real data  |         1.8% |           0.7% |          1.8% |               n.a. |           n.a. |                            n.a. |
>
> We can see from the table that the 16-gram repetition is nearly 0 for both 200B-scale and 1T-scale. This means the synthesized documents indeed generate novel content that the seed doesn’t already see.
>
> > Statistics about the length of synthetically generated documents?
>
> Thanks for the question. We report the experiment length at the 200B-scale and contrast it with real data.
> | Statistic |Real data|200B-scale generated|
> | :---      |    ---: |               ---: |
> | Mean      |  1338.2 |              795.0 |
> | Std       |  1096.7 |              623.8 |
> | 25%       |   560.8 |              413.8 |
> | 50%       |   918.0 |              667.0 |
> | 75%       |  1801.2 |             1012.0 |
> We can see that the synthetic data are systematically shorter compared with the real data.

---

### Meta-Review · Area_Chair_6g3i · 2026-01-06

**Summary:**

This paper presents a method to generate synthetic data for pretraining in constrained data settings to explore relations between documents. Reviewers find the proposed method is a creative way to address the issue of the data “scaling wall”. While several questions and concerns are raised based on the initial submission, they include concerns about generalization to larger models, concerns regarding high non-factual error rates, requests for experiments on random pair ablation, and requests for experiments to verify data novelty, etc. I believe the detailed response largely addresses the concerns and questions. While some may not be fully addressed, e.g., “With a larger model, it might be necessary to conduct fair comparative experiments on a larger amount of data” from Reviewer 9CYh, this may require a significant amount of resources, which is hard to achieve during the rebuttal period. Overall, I find this is an interesting paper with a solid set of experiments after incorporating all the rebuttal details.

**Reviewer Concerns:**

Reviewer Vkk4 has a few questions regarding reporting the known biases set up in pretraining, the unclear improvement brought by the data synthesizer, quantitative results on the 16-gram metric, and statistics about the length of generated documents. In the rebuttal, I believe all questions and concerns are well addressed. For reviewer s1yS, the major concerns are the high non-factual error rate, the reason for using the Qwen3-0.6B embedding model, and the false claim of the learned relationship as “causality.” In the rebuttal, the authors provide additional experiments at the 1T scale with a 6B-parameter model. For the remaining two questions, the authors also provide reasonable answers for clarification. For reviewer 9CYh, in the discussion, the reviewer mentions that he/she still has concerns and would like to conduct fair comparative experiments on a larger amount of data with a larger model. While it is unfortunate that the discussion is not complete, after the authors provide further information. This further clarification sounds reasonable to me. Also, given the high compute cost due to the nature of pretraining, it is understandable that not all requested experiments can be completed. For reviewer 6oUP, the authors provide a detailed response, including comparisons with random documents, and also clarify additional questions. I believe the detailed response largely addresses the reviewers’ concerns.

**Reviewer Scores:**

I think Reviewer Vkk4 may raise the rating to 8. Reviewer s1yS may raise it to 6. Reviewer 9CYh said he/she would remain unchanged at 4; however, the discussion was not complete. The provided information sounds reasonable to me, and the reviewer may also raise it to 6. Reviewer 6oUP may also raise it to 6, since the concerns are largely addressed.

---

### Decision · Program_Chairs · 2026-01-26

Accept (Poster)